# Biorefinery Processing of Waste to Supply Cost-Effective and Sustainable Inputs for Two-Stage Microalgal Cultivation

Pierre C. Wensel [1,*], Mahesh Bule [1], Allan Gao [1], Manuel Raul Pelaez-Samaniego [1,2], Liang Yu [1], William Hiscox [3], Gregory L. Helms [3], William C. Davis [4], Helmut Kirchhoff [5], Manuel Garcia-Perez [1] and Shulin Chen [1,*]

[1] Department of Biological Systems Engineering, Washington State University, Pullman, WA 99164, USA; mahesh.uict@gmail.com (M.B.); allan.gao@gmail.com (A.G.); manuel.pelaez@ucuenca.edu.ec (M.R.P.-S.); yuliang08@wsu.edu (L.Y.); mgarcia-perez@wsu.edu (M.G.-P.)

[2] Department of Applied Chemistry and Systems of Production, Universidad de Cuenca, Cuenca 010158, Ecuador

[3] Center for NMR Spectroscopy, Washington State University, Pullman, WA 99164, USA; hiscox@wsu.edu (W.H.); greg_helms@wsu.edu (G.L.H.)

[4] College of Veterinary Medicine, Washington State University, Pullman, WA 99164, USA; davisw@wsu.edu

[5] Institute of Biological Chemistry, Washington State University, Pullman, WA 99164, USA; kirchhh@wsu.edu

[*] Correspondence: pierrewensel@gmail.com (P.C.W.); chens@wsu.edu (S.C.); Tel.: +1-509-335-3743 (S.C.); Fax: +1-509-335-2722 (S.C.)

**Featured Application: This described technology can be applied for cost-effective manufacturing of food, feed, fuel, therapeutics, and bioplastics via algal cultivation.**

**Abstract:** Overcoming obstacles to commercialization of algal-based processes for biofuels and co-products requires not just piecemeal incremental improvements, but rather a comprehensive and fundamental re-consideration starting with the selected algae and its associated cultivation, harvesting, biomass conversion, and refinement. A novel two-stage process designed to address challenges of mass outdoor microalgal cultivation for biofuels and co-products was previously demonstrated using an oleaginous, haloalkaline-tolerant, and multi-trophic green *Chlorella vulgaris*. ALP2 from a soda lake. This involved cultivating the microalgae in a fermenter heterotrophically or photobioreactor mixotrophically (first-stage) to rapidly obtain high cell densities and inoculate an open-pond phototrophic culture (second-stage) featuring high levels of $NaHCO_3$, pH, and salinity. An improved two-stage cultivation that instead sustainably used as more cheap and sustainable inputs the organic carbon, nitrogen, and phosphorous from fractionation of waste was here demonstrated in a small-scale biorefinery process. The first cultivation stage consisted of two simultaneous batch flask cultures featuring (1) mixotrophic cell productivity of $7.25 \times 10^7$ cells $mL^{-1}$ $day^{-1}$ on $BG-11_0$ medium supplemented with 1.587 g $L^{-1}$ urea and an enzymatic hydrolysate of pre-treated (torrefaction + grinding + ozonolysis + soaking ammonia) wheat-straw that corresponded to 10 g $L^{-1}$ glucose, and (2) mixotrophic cell productivity of $2.25 \times 10^7$ cells $mL^{-1}$ $day^{-1}$ on $BG-11_0$ medium supplemented with 1.587 g $L^{-1}$ urea and a purified and de-toxified condensate of pre-treated (torrefaction + grinding) wheat straw that corresponded to 0.350 g $L^{-1}$ of potassium acetate. The second cultivation stage featured [1]H NMR-determined phototrophic lipid productivity of 0.045 g triacylglycerides (TAG) $L^{-1}$ $day^{-1}$ on $BG-11_0$ medium supplemented with 16.8 g $L^{-1}$ $NaHCO_3$ and fed batch-added 22% (*v/v*) anaerobically digested food waste effluent at HCl-mediated pH 9.

**Keywords:** biorefinery; algae; anaerobic digestion; torrefaction; enzyme hydrolysis; food and lignocellulosic waste

## 1. Introduction

Commercialization of algal-based processes for biofuels and co-products is constrained upstream during cultivation by low productivity, contamination by invasive species, unsustainable and inefficient supply of nutrients (e.g., nitrogen, sulfur, and phosphorous),

inorganic carbon, and water, exposure to environmental factors, and limited available land acreage for inoculum and subsequent large-scale cultures. It is also constrained downstream by costly and inefficient harvesting, cell disruption, product extraction, thermochemical conversion, hydrotreatment, and upgrading. A novel two-stage cultivation process was previously proposed to address various challenges of mass microalgal cultivation for bio-fuels and co-products and demonstrated using an oleaginous, haloalkaline-tolerant, and multi-trophic green *Chlorella* sp. ALP2 isolated from a soda lake [1]. This involved cultivating the microalgae in a fermenter heterotrophically or photobioreactor mixotrophically on organic carbon (first-stage) to rapidly obtain high cell densities and inoculate an open-pond phototrophic culture (second-stage) featuring high levels of $NaHCO_3$, pH, and salinity. The combination of high cell densities and extreme haloalkaline conditions in the second phototrophic stage could then minimize residence time and limit contamination [2], facilitate auto-flocculation harvesting [3], efficiently supply inorganic carbon [4], and enhance neutral lipid accumulation [5]. This two-stage process can be further improved upon with sustainable supplies of organic carbon for the first stage and nitrogen and phosphorous nutrients for the second stage.

Organic carbon for the first-stage enclosed microalgal cultures can be derived from hydrolysates and/or condensates from processed ligno-cellulosic agricultural and/or forestry wastes. For instance, a glucose-containing enzymatic hydrolysate from high-temperature ethanol-treated rice-straw and an acetic acid-containing condensate from fast-pyrolyzed softwood were previously used to cultivate the microalgae *Chlorella pyrenoidosa* [6] and *Chlamydomonas reinhardtii*, respectively [7]. Hydrolysates from treated ligno-cellulosic biomass are advantageous over those resulting from starch and juice of agricultural crops such as Jerusalem artichoke, cassava, and sugarcane [6]. Namely, they contain both $C_6$ and $C_5$ sugars, avoid the "food vs. fuel" controversy, and are more cost-effective and higher-yielding than purchased glucose, which, in turn, was previously shown to be the highest-yielding carbon substrate for the *Chlorella vulgaris* ALP2 microalgae, particularly under mixotrophic conditions [1]. Alternatively, condensates from thermochemically processing lignocellulosic biomass, i.e., fast-pyrolysis, represent a way of removing the carboxylic acids and levoglucosan used for microalgal [7] and yeast cultivation [8] while simultaneously stabilizing pH, reducing corrosivity, and increasing heating-value of an intended bio-oil intermediary product that is later refined to fuel (i.e., green gasoline) or other chemicals.

Despite these improvements, the conventional dilute sulfuric acid approach for hydrolysates and the fast-pyrolysis approach for condensates both have limitations. First, they require relatively expensive capital equipment for high-temperature and/or high-pressure operating conditions [9]. Second, such extreme conditions result in high levels of compounds in the hydrolysate and/or condensate that may inhibit microalgal growth [10]. Third, they fail to adequately address major logistical challenges and high costs (i.e., 30% of feedstock costs) associated with initially harvesting, transporting, storing, and grinding waste ligno-cellulosic biomass feedstock to uniform particle size for greater process efficiency prior to pre-treatment at a biorefinery [11]. Fourth, the selected lignocellulosic biomass may not be the most suitable. For instance, pyrolyzed forestry residues (i.e., softwood and hardwood) will yield less acetic acid and be less available, particularly during dark and cold winters when either indoor microalgal hetero-/mixotrophic cultivation or firewood combustion are implemented, compared to the very abundant, grassy agricultural crop residues (i.e., wheat-straw) after fall harvests [12,13]. Such by-products comprised up to 50% by weight of the world's 2 billion t year$^{-1}$ cereal crop production in 2011 [6].

Rather than incorporating these aforementioned lignocellulosic biomass treatment methods to supply organic carbon for the first of a two-stage microalgal cultivation process, a novel, more cost-effective, hybrid, thermo-biochemical pre-treatment method involving torrefaction of wheat-straw and subsequent steps was therefore devised. Compared to fast-pyrolysis, torrefaction is a lower-temperature (200–300 °C) and self-sustaining thermo-chemical process occurring in an inert, $O_2$-less atmosphere [11] that can be conducted in

relatively simpler reactors like those previously used by Tawainese farmers [14]. It results not only in a drier, more brittle, and hydrophobic solid biomass to alleviate feedstock logistical issues [15], but also in high-heating value gases (CO, $CH_4$) combusted to run it [16], and a liquid condensate containing a mixture of water and carboxylic acids [11]. These torrefaction-derived acids arise mostly from the thermochemical degradation of hemi-cellulose as a by-product, while the pyrolysis-derived acids and phenolic growth inhibitors arise mostly from the lignin depolymerization as an intended bio-oil product [11]. The liquid condensate remaining after torrefaction, requiring less extensive purification due to the lower temperatures, can potentially supply organic carbon (acetate) to a first-stage microalgal culture. In this way, a microalgae that is unable to uptake and/or metabolize directly $C_5$ sugars (i.e., xylose, mannose, ribose) in hydrolysates can use instead the acetic acid in the condensate obtained from the same hemicellulose fraction. The solid biomass remaining after torrefaction can also potentially be further pre-treated with ozonolysis ($O_3$) and soaking in recyclable ammonia, enzymatically hydrolyzed, and purified to supply organic carbon ($C_6$ sugar glucose) to another first-stage microalgal culture. Compared to the dilute sulfuric acid method, these novel pre-treatment steps involving $O_3/NH_4$ to generate hydrolysate were previously shown to require less operation time and milder operating conditions, yielding higher levels of $C_6$ sugars and lower levels of phenolic inhibitors [10].

Nitrogen and phosphorous nutrients for second-stage, large-scale outdoor open-pond microalgal cultures can be sustainably supplied by effluents from the integrated, bio-gas generating, anaerobic-digestion (AD) of waste. AD using mesophilic or thermophilic bacteria has been widely used instead of energy-intensive aeration-based activated sludge processes to partially treat swine, poultry, and dairy manure waste from farm concentrated animal feeding operations by reducing organic matter and waste volume via fermentative degradation of organic constituents and prevent eutrophication upon its discharge into water bodies [17]. Effluents' unmetabolized inorganic nitrogen (i.e., ammonia) can be used by the microalgae [18], and biogas (i.e., typically a mixture of $CH_4$, $CO_2$, $H_2S$, and $H_2$) can be purified and used for on-site electrical energy to power compressors, pumps, and harvesting centrifuges. Microalgal photosynthesis during cultivation on such waste effluents evolves dissolved $O_2$ to kill contaminating anaerobes. It is also more metabolically efficient [19] and cost-effective [20] than that using $NaNO_3$, which represents the most expensive (750–950 USD $t^{-1}$) and concentrated (1.500 g $L^{-1}$) component of the traditional BG-11 growth medium devoid of vitamins. For instance, based on flux balance analysis, nitrates are more metabolically expensive than ammonia because phototrophs, such as *Synechocystis PCC6803*, divert and expend 9 moles of NADPH and ATP to convert 1 mole of $NaNO_3$ into 1 mole of preferred ammonia [19,21].

However, despite these advances, batch addition of effluents derived from anaerobically digesting animal waste manure to open-ponds also has limitations: first, manure effluents and even their dilutions tend to be turpid and can block transmission of sunlight to microalgae cultivated in second-stage outdoor open-ponds [18], which has been shown to significantly influence productivity [22]. Second, economic losses and pollution due to effluent ammonia volatilization [23] during hot summers would likely be higher in the large-scale second-stage open-ponds proposed here with high surface areas, temperatures, pH, and wind-speeds. Third, the combination of the UV wavelength of polychromatic light (i.e., outdoor sunlight) and high concentrations of protonated ammonia in batch-added AD effluent are also reportedly more toxic to the PSII photosynthetic apparatus of phototrophs at the proposed elevated pH, where uncontrollable, diffusive, non-active transport across the cell boundary occurs [24].

Rather than incorporating the aforementioned batch addition of AD manure effluents to supply nitrogen and phosphorous for the second of a two-stage microalgal cultivation process, a novel method involving fed-batch addition of AD effluent derived from food waste was developed. Food waste from municipalities and food processing facilities represents another alternative source of AD effluents. Food waste diversion from landfills is

considered as a way to increase landfill capacity and reduce methane emissions, with a large potential energy content amounting to 2% of total U.S. energy consumption [25]. Compared to the turbid dairy manure effluents, light yellow–brown effluent from starchy food-waste additionally represents greater microalgal open-pond light transmission. An effluent fed-batch strategy at high pH in the proposed second stage culture should employ an exponentially increasing mass flow [26]. This would need to balance (1) the rate of effluent supplementation required for both sufficient growth and terminal nitrogen starvation-induced lipid accumulation, with (2) the rate of microalgal ammonia consumption and (3) the rate of high pH-induced volatilization.

In this study, proof of concept was established by cultivating for the first time an oleaginous, multi-trophic, haloalkaline-tolerant microalgae in an improved two-stage process using integrated, sustainable supplies of organic carbon from wheat-straw for mixotrophic first-stage and nitrogen and phosphorous from food waste for the phototrophic second stage. Carbon and nutrient sources were additionally analyzed in various conditions.

## 2. Materials and Methods

### 2.1. Microalgal Strain for Growth Experiments

A multi-trophic (hetero-/mixo-/phototrophic), oleaginous, halo-alkaliphilic green microalgae *Chlorella vulgaris* ALP2 was isolated and characterized as previously described [1].

### 2.2. Torrefaction of Wheat-Straw

Wheat straw (*Triticum aestivum*) was obtained in dried farm bails (Grange Supply Co., Pullman, WA, USA), ground using a hammer mill, and sieved to 42–60 mesh size with two screens. Approximately 224 g and 570 g of dry wheat straw were torrefied in a previously described auger reactor [27] at 280 °C and 300 °C, respectively, with 17 min residence time and inert $N_2$ gas (10 L min$^{-1}$). The process resulted, respectively, in 209.63 g and 383.84 g of solid torrefied biomass (mass yields of 93.6% and 67.3%, respectively) that were collected in a closed metal cylinder, gas products evacuated by a vacuum pump, and 12.1 mL and 147 mL, respectively, of a liquid condensate (50 mL dark brown primary condensate). The torrefaction process and the characterization of all materials (see details in following sub-sections) were conducted in duplicates. To simulate a biorefinery's biomass inventory, all of the non-torrefied and torrefied solid biomass was stored for 14 days in a dry room at 23 °C, and the highly acidic and unstable condensates were immediately frozen at −20 °C for future use.

### 2.3. Analysis of Torrefied and Non-Torrefied Wheat-Straw and Torrefaction Condensate

The torrefied and non-torrefied solid biomass were comparatively analyzed using scanning electron microscopy (SEM) imaging, pyrolysis-gas chromatography/mass spectrometry (Py-GC-MS), Fourier-transform infrared spectroscopy (FTIR) with attenuated reflectance (ATR), and thermogravimetric analysis (TGA), as previously described [28]. The liquid torrefaction condensates were analyzed using gas chromatography (GC), acidity (using a pH meter), and water content (Karl–Fisher moisture testing). The thermal behavior of raw and 300 °C-torrefied wheat-straw under different heating rates was evaluated through TGA (Thermogravimetric Analysis), using a TGA/SDTA 85/e TGA analyzer (Mettler-Toledo, Columbus, OH, USA) by heating a typical sample mass of approximately 10 mg placed in 70 μL alumina/ceramic-based sample holders of a TSO801RO Sample Robot auto-sampler and balance in a purge of nitrogen (50 mL min$^{-1}$), at a pre-programmed linear heating rate of 10, 20, 30, and 40 K min$^{-1}$. The final temperature was 873.5 K, with a holding time of 15 min. For each heating rate, the % converted data were normalized to reflect that heating up to 150 °C was necessary to eliminate the residual water moisture content of the torrefied wheat-straw. Conversion was calculated following Equation (1).

$$\alpha = \frac{(m_0 - m)}{(m_0 - m_\infty)} \tag{1}$$

where $m_0$ = 100%, $m_\infty$ is the last corrected TG% value and DTG was calculated using Equation (2).

$$DTG = \frac{(m_{n+1} - m_n)}{dt}$$ (2)

where $m_{n+1}$ refers to the measured mass at a specific time ($t$) and $m_n$ refers to the previous mass measurement, and $dt$ refers to the time increase from one measurement to the next one.

For each of the four heating rates, the sample temperatures and $\ln(d\alpha/dt)$ values corresponding to conversions of 0.05 increments in the range of 0.05–0.90 were determined. A first-order reaction rate law with a temperature-dependent rate law constant $k_A$ expressed by the Arrhenius Equation (3) was assumed to describe the resulting homogeneous primary thermochemical reactions with negligible mass and heat transfer.

$$k_A(T) = A e^{-\frac{E_a}{RT}}$$ (3)

where $A$—the pre-exponential factor or frequency factor, $E_a$—activation energy (J mol$^{-1}$), $T$—absolute temperature (K), and $R$—gas constant (8.314 J (mol K)$^{-1}$). Milling of the wheat-straw to particle size between 42 and 60 mesh and use of an inert nitrogen carrier gas ensured rapid heat and mass transfer and avoidance of undesirable secondary thermochemical reactions. The isoconversional Friedman method [29], which assumes that the reaction rate ($d\alpha/dt$) at a constant conversion is only a function of temperature, was applied to thermogravimetric data. This was used to determine kinetic parameters for a first-order reaction rate law assumed to describe homogeneous primary thermochemical reactions with negligible mass and heat transfer. A plot of $\ln(d\alpha/dt)$ vs. $1000/T$ (K) for each conversion was generated. The activation energies were calculated from the slopes of each relatively straight line as per Equation (4).

$$slope = \frac{-E_a}{R}$$ (4)

The Arrhenius pre-exponential factors were calculated from the $x$-intercepts of these relatively straight lines following Equation (5), where a reaction order $n$ of 1 was assumed.

$$\ln A = x - \text{intercept} - n \ln(1 - \alpha)$$ (5)

The ASTM E1641-04 integral method was alternatively used. The vaporized products from primary reactions of fast pyrolysis of 280 °C-torrefied and non-torrefied wheat straw were also comparatively obtained and analyzed on a Py-GC-MS instrument (Agilent, Santa Clara, CA, USA) with a calibrated oven temperature of 500 °C and heating rate of 800 °C min$^{-1}$. For this, samples of 1.54 mg and 0.620 mg were weighed for 280 °C and 300 °C-torrefied and raw wheat-straw, respectively, using the aforementioned TGA instrument.

### 2.4. Preparation and Analysis of Wheat Straw Hydrolysate

To generate hydrolysate, 10 g of 280 °C-torrefied and non-torrefied biomass were soaked at 10% ($w/v$) overnight in de-ionized water and ozone-lyzed in a metallic batch reactor as previously described [30], in 2.5 g-increments for 30 min with air/$O_3$ flow-rate of 5 L min$^{-1}$. Approximately 7.63 g of biomass was then used for subsequent pretreatment. This involved soaking at 10% ($w/v$) in an aqueous 28–30% ($w/w$) NH$_4$OH solution (JTB-9721-03) and incubating at 50 °C for 12 h with no agitation in 1 L screw-cap Pyrex solution bottles. The biomass was then washed thoroughly with 0.8 L deionized water through a vacuum-pressurized Whatman filter until neutral pH 7.0 in the permeate was reached. The resulting filter-cake (g) was dried at 50 °C for 8 h. Then, 4 g of it was enzymatically hydrolyzed at 50 °C in a capped flask in an Gyromax 747 orbital incubator shaker (Amerex Instruments, Inc., Concord, CA, USA) for 72 h. This occurred at 4% ($w/v$) solid loading in

0.050 M sodium acetate buffer (pH 4.8) supplemented with 8 mL of solution containing 2% sodium azide with 30 FPU g$^{-1}$ of cellulase (Novozymes NS 50013) and 30 CBU g$^{-1}$ of β-glycosidase (Novozymes NS 50010). Enzymatic hydrolysis was terminated by boiling the hydrolysate in a hot water bath for 5 min and centrifuging for 10 min at 3000 rpm ($\times g$) to remove residual insoluble lignin particulates. About 48 mL of light-brown hydrolysate was then decolorized and detoxified by adding 2.4 g Darco 20–40 mesh activated charcoal (Sigma-Aldrich, St. Louis, MI, USA) at a 5% (*w/v*) loading, briefly vortexing, and incubating for 12 h at 4 °C. The activated charcoal used for de-toxifying non-torrefied hydrolysate was also vigorously vortexed and incubated for 16 h at 4 °C in acetone and methanol solvent in 1 mL Eppendorf tubes and then PTFE 0.2 um disc-filtered away to remove coloration compounds for analysis on GC-MS as previously described [31]. The resulting black slurry was centrifuged, and grey supernatant was vacuum-filtered with Whatman paper. The clear permeate was then neutralized to pH 7 via addition of 1 M NaOH, sterilized with 0.2 μm disc-filter for subsequent mixotrophic microalgal cultivation, and analyzed for sugar content.

Additional non-detoxified hydrolysates of both non-torrefied and torrefied biomass derived instead by 2% solid loading and using instead the 0.050 M sodium citrate buffer specified in an NREL standardized method [32] were also comparatively analyzed for sugar content and ability to be used for microalgal cultivation. The final acetate-buffering and de-toxification steps to generate the hydrolysate used for microalgal cultivation were devised after preliminarily observing that citrate buffer did not permit growth consistent with the literature and that 33% and 50% dilutions of the brown hydrolysate with deionized water allow progressively greater microalgal growth.

### 2.5. Preparation and Analysis of Purified Wheat Straw Torrefaction Condensate

Of the 50 mL of thawed condensate (bio-oil), 4 mL was set aside for Karl–Fisher moisture testing. In two separate tubes, about 23 mL condensate was diluted with 11 mL cold de-ionized water and agitated horizontally in 50 mL plastic tubes on an orbital flask for 1 h to precipitate out phenolic-containing resins that adhered to tube walls. Each supernatant was then neutralized with 6 mL of 10 N KOH base addition from pH 2.54 to 7.00 to prevent volatilization of acids and then centrifuged at 4000 rpm ($\times g$) for 10 min to separate away any residual solids. The supernatants were extracted 1:1 (*v/v*) with ethyl acetate as the top layer. These were then distilled at vacuum pressure (~30 torr) at 60 °C for 20 min with 30 rpm rotation with cooling water and a three-bulb collector of a Rota-Vapor R-100 distillation unit ( Buchi LaborTechnik AG, Flawil, Germany). A total of 70 mL was collected in the bottom bulb. The clear liquid that boiled away and condensed in the middle second bulb collector was later analyzed for $^1$H-NMR analysis. About 45 mL of the condensate remaining in the bottom-bulb was then incubated overnight at 4 °C at 5% (*w/v*) with a combination of 3.26 g regular and 3.26 g O$_3$-treated activated-carbon. O$_3$-treatment was carried out as previously described [30] to add repulsive negatively charged surface-moieties which prevent adsorption losses of valuable acetate ions. The liquid condensate was then filtered with Whatman-paper under a vacuum and then using 0.45 μm and 0.2 μm disc filters, yielding a 34 mL of a yellow and clarified condensate. A 100$\times$ dilution of this supported algal growth but with a long lag-phase. Therefore, 25 mL of the condensate underwent the vacuum distillation procedure again but this time for an additional 1.5 h. About 3.5 mL of dark-orange, viscous distillate containing non-volatilized carboxylate salts that were retained in the first bulb collector was allowed to cool and re-suspended back to 25 mL with deionized water to enable collection and filtration. About 9.5 mL and 4 mL of clear liquid were also collected in the large top bulb and center mid-bulb, respectively. Carboxylic acid and ethanol content of the final purified and de-toxified condensate was measured by withdrawing a small sample, lowering its pH to 2.5 with HCl acid, and loaded onto a GC as previously described [31].

### 2.6. Preparation of Effluent and Biogas from Anaerobic Digestion of Food, Manure, and Torrefaction Condensate

Food waste effluent was obtained from the 7-day processing of American-style food waste from a Washington State University (WSU) cafeteria in a system comprised of a high-solids anaerobic digester with recycling seed (SADRS) for hydrolysis and an Up-Flow Anaerobic Sludge Blanket (UASB) reactor for methanogenesis described elsewhere [33]. Approximately 20 L day$^{-1}$ of biogas on average comprising 62% $CH_4$, 37% $CO_2$, at 0.29 L $CH_4$ g$^{-1}$ volatile solids food waste and 19.2 L day$^{-1}$ of food waste effluent was generated. Flush dairy-manure waste effluent was derived from the Sequential Batch Reactor (SBR) as previously reported [34]. To initially test if effluent alkalinity and salinity sufficiently prevented ammonia volatilization losses at high temperatures and to sterilize preliminary comparative flask cultures, both effluents were autoclaved for 25 min at 121 °C and 101.1 kPa gauge pressure and subsequently analyzed with an alkalinity meter and the Kjeldahl digestion method [35]. Samples from cultures involving manure effluent were also analyzed on a FACSCalibur flow cytometer (Becton Dickinson Immunocytometry Systems, San Jose, CA, USA) for scattering and fluorescence as previously described [1]. For subsequent microalgal growth experiments, both effluents were instead briefly vortexed to remove adsorbed N and P species from suspended particles and sterilized under vacuum with 0.2 μm filtration (Millipore-Sigma, St. Louis, MO, USA) to remove contaminating bacteria and, in the case of manure effluent, suspended particles that obfuscate microalgal dry cell weight and UV-Vis spectrophometric measurements.

### 2.7. Statistical Analysis

Preliminary heterotrophic and phototrophic cultivation experiments were performed using balanced duplicate replication, randomization, and negative controls. Data were statistically analyzed by one-way analysis of variance (ANOVA). Statistical significance was evaluated by estimation of the descriptive level ($p$), where results were considered statistically significant when $p < 0.05$ ($\alpha = 0.05$, confidence level of 95%).

### 2.8. Biomass, Lipid, and Extracellular Concentration Measurements

Dry cell weight (*DCW*), chlorophyll a/b and carotenoid content, extracellular pH, light intensity, optical density (*OD*) were measured as previously described [10]. Absorbance maxima at $OD_{680}$ corresponding to algal chlorophyll was not found to be correlated to absorbance minima at $OD_{750}$. Separate standard curves were generated for heterotrophic and phototrophic cultures because $OD_{680}$ depends on dynamic microalgal optical properties (i.e., Mie-scattering and Beer's Law absorption extinction cross-sections, which are related to chlorophyll LHCII antenna size and cellular dimensions) that, in turn, are dependent on the incident light intensity [22]. A standard curve correlating heterotrophic $OD_{680}$ and *DCW* for ALP2 was therefore developed with linear regression as follows:

$$DCW(\text{g/L}) = OD_{680} \times 0.4007 - 0.0172, \; R^2 = 0.9734 \tag{6}$$

Similarly, a standard curve correlating phototrophic $OD_{680}$ and *DCW* for ALP2 was developed:

$$DCW(\text{g/L}) = OD_{680} \times 0.3713 + 0.1617, \; R^2 = 0.9068 \tag{7}$$

Specific growth rates during the exponential growth phase between initial ($t_1$) and final ($t_2$) time points were calculated as previously described [10]. Total carbon and inorganic carbon in supernatant were measured using a TOC-5000 analyzer (Shimadzu, Kyoto, Japan). Total nitrogen was measured using a spectrometer and total high-range (10–150 mg L$^{-1}$) colorimetric reagent Test N Tube kits (Hach Company, Loveland, CO, USA). An un-inoculated flask containing BG-11$_0$ medium, adjusted to pH 9 and supplemented with 2.5% (*v*/*v*) effluent and 17.0 g L$^{-1}$ $NaHCO_3$, was agitated to determine its ammonia volatilization rate, based on periodic measurement of total nitrogen in the

medium. Neutral lipid content was assessed for the majority of ALP2 cultures via $^1$H NMR, as previously described [10]. Lipid content and fatty-acid profiles for ALP2 cultures grown heterotrophically at 28 °C and 21 °C at 10 g L$^{-1}$ and 20 g L$^{-1}$ glucose, or phototrophically on food waste effluent, were assessed by a previously described FAME GC-based procedure [23].

### 2.9. Evaluation of Second Stage Cultivation Conditions

ALP2 was centrifuged, washed once in PBS, and inoculated at an $OD_{680}$ = 0.035 from 14-day maintenance cultures into 250 mL flasks containing 100 mL of BG-11$_0$ [36] media devoid of KH$_2$PO$_4$ and supplemented with different anaerobically digested food waste effluent concentrations (5%, 10%, 15%, 22% (*v/v*)) and different NaHCO$_3$ concentrations (8.4 g L$^{-1}$, 16.8 g L$^{-1}$, 25.2 g L$^{-1}$, 33.6 g L$^{-1}$) with pH either un-adjusted or daily adjusted to 9.0 via HCl acid. The flasks were agitated at 150 rpm on orbital shaker under 75 μmole m$^{-2}$ s$^{-1}$ PAR light intensity delivered by polychromatic fluorescent light bulbs overhead. These cultures were monitored for $OD_{680}$, pH, flow cytometric BODIPY-488 stain-based lipid content (1.5 mL day10), chlorophyll-fluorescence (1.5 mL day 4), O$_2$-evolution, and respiration. Finally, 50 mL and 20 mL samples were withdrawn for measuring biomass dry cell weight and FAME-GC-derived lipid content, respectively.

### 2.10. Demonstration of Two-Stage Cultivation

A two-stage cultivation process was demonstrated using the organic carbon, nitrogen, and phosphorous from biological waste. For two simultaneous first-stage mixotrophic cultures, ALP2 from maintenance culture was centrifuged, washed twice in PBS, and inoculated at an $OD_{680}$ = 0.50 into two foam-capped 50 mL flasks. These two flask cultures were agitated at 150 rpm on orbital shakers under 75 μmole m$^{-2}$s$^{-1}$ light intensity from overhead polychromatic fluorescent bulbs and contained 25–30 mL of 0.2 μm-filtered BG-11$_0$ adjusted to pH = 7.0 with 1 M NaOH, supplemented with 1.587 g L$^{-1}$ urea and either (1) torrefied wheat-straw hydrolysate diluted to 10 g L$^{-1}$ glucose or (2) torrefied-wheat-straw condensate diluted to 0.35 g L$^{-1}$ potassium acetate. Such dilutions of condensate and hydrolysate further ensured lower concentration of inhibitory compounds (i.e., phenolics, furans (HMF and aldehydes)) for microalgal cultivation. (It contained 25 mL of modified BG-11 medium supplemented with 0.250 m L of purified condensate representing a dilution to approximately 0.35 g L$^{-1}$ potassium acetate). These were cultivated mixotrophically to the near-stationary phase. The cells from these two cultures were then washed once with PBS to remove any residual nitrogen, organic carbon, and phosphate. They were then inoculated at $OD_{680}$ = 0.250 each for a combined total of $OD_{680}$ = 0.500 corresponding to a cell density of 2.5 × 10$^6$ cells mL$^{-1}$ into a second-stage, foam-capped 250 mL Erlenmeyer flask. This second-stage flask culture was cultivated phototrophically at 22 °C under 75 μmoles m$^{-2}$s$^{-1}$ photon flux density from overhead polychromatic fluorescent light bulb, agitated at 150 rpm on orbital shaker, and contained 100 mL of BG-11$_0$ medium devoid of KH$_2$PO$_4$ and supplemented with 16.8 g L$^{-1}$ NaHCO$_3$ and 2.5% (*v/v*) of 0.2 μm-filtered anaerobically digested food waste effluent, at a pH 10 that was daily adjusted initially with 0.5 M HCl acetic acid. An exponential fed-batch strategy [37,38] for ammonia-containing food waste effluent was subsequently used and expressed as follows Equation (8):

$$m_t = m_0 e^{kt} \tag{8}$$

### 2.11. Intracellular and Extracellular Cultivation Measurements

Optical density (*OD*) at 680 nm, dry cell weight from 0.5 M NH$_3$HCO$_3$-washed culture samples, haemocytometric cell density, chlorophyll a/b content, specific growth rate, alkalinity, pH of culture media and lake water, light intensity, total carbon and inorganic carbon, total nitrogen and ammonia nitrogen, and were measured and calculated as previously described [1]. Total phosphate PO$_4$$^{3-}$ and Chemical Oxygen Demand (COD) were measured as reported [1]. Alkalinity of effluent and culture supernatants was measured using

a T-50 Rondolino analyzer (Mettler Toledo, Columbus, OH, USA)as reported as mg $L^{-1}$ of $CaCO_3$. Total $NH_4$-N composition of effluent and culture supernatants was measured using a Tecator 2300 KJELTEC analyzer (Tecator, Apeldoorn, The Netherlands). A standard curve correlating phototrophic $OD_{680}$ and *DCW* was developed with linear regression following Equation (9).

$$OD_{680} = DCW(\text{g/L}) \times 0.4007 - 0.0172, \text{ R}^2 = 0.9734 \qquad (9)$$

Absolute intracellular neutral lipid content was assessed using liquid-state $^1$H NMR and a FAME GC-FID procedure, as previously described [1]. Relative intracellular neutral lipid content and the relative densities of chlorophyll-fluorescent microalgal ALP2 cell populations and non-fluorescing anaerobic digestion effluent debris were obtained using a benchtop FACSCalibur flow cytometer equipped with CellQuest software (Becton Diskinson Immunocytometry Systems, San Jose, CA, USA) as previously described [1]. BODIPY 488 -staining of once- PBS-washed culture samples was performed using a previously described procedure whereby 0.5 mL of ALP2 microalgal culture was stained directly with 1 μM BODIPY-488 in TFE solvent and incubated for 5 min. These and non-stained auto-fluorescent negative control samples were then acquired with flow cytometrically. The ratio of FL1 of stained to that of the unstained sample gave the normalized fluorescences, and this was used to compare the ALP2 relative neutral lipid content at different conditions. Extracellular content of $C_5$ (i.e., xylose, mannose, ribose) and $C_6$ (glucose, galactose) monomer sugars from first-stage mixotrophic cultures and in hydrolysates before and after detoxification was analyzed with a Dionex Ion-Exchange-HPLC (Agilent, Santa Clara, CA, USA) equipped with an amperemeter/FID detector as previously described [1].

## 3. Results and Discussion

### 3.1. Analysis of Raw and Torrefied Wheat Straw

In this study, torrefaction of wheat-straw was performed as the first pre-treatment step to prepare both an enzymatic hydrolysate and a purified liquid condensate supplying organic carbon to first-stage microalgal cultures as part of a previously described two-stage process. Raw lignocellulosic biomass like wheat-straw is characterized by high moisture content, relatively low energy density and larger volume, and hygroscopic behavior. These properties make it expensive to transport, store, and grind into small particles of uniform size. To supply a feedstock for continuous year-round biorefinery operation, biomass has to be collected from large and often distant areas and hauled either to local storage facilities or to a refinery where it would be stored until conversion. Unfavorable hydroscopic physical properties of biomass dictate using large, uneconomical, storage facilities. Storing large amounts of wet biomass will further increase expenses through the high rate of dry matter loss due to microbial activity and the hazard of self-heating/combustion.

Torrefaction is a thermochemical process previously shown to upgrade lignocellulosic biomass feedstock as follows: first, torrefied biomass is more homogeneous with a controlled particle size distribution that simplifies a biorefinery's feedstock quality control. Second, it is more brittle [39] and easier to grind based on higher Hardgrove Grindability Index. Third, it is more energetically dense, making it more convenient to transport. Fourth, it is less hydrated, making it easier to store by reducing the risk of spoilage and barn fires caused by microbially derived methane emissions. Fifth, it has a lower O/C ratio and higher calorific value, well below but nonetheless approaching that of coal used in gasification and/or combustion processes. The attendant capital and operating costs, as well as minute conversion losses, of an additional torrefaction unit of operation either off-site or at the biorefinery were demonstrated to be offset by savings elsewhere, such as entrained flow gasification, small scale combustion using pellets and co-firing in pulverized coal fired power stations. That even contracted farmers who supply the biorefinery feedstock include a torrefaction unit in their operations in the future seems possible considering that a small, inexpensive, modular prototype torrefaction system was devised and successfully

implemented for farmed camper wood, bamboo, and rice straw as part of a "2010 Project on the Promotion of Green Community in Chiayi County" in Taiwan [14].

During torrefaction, the lignin, hemicellulose, and cellulose polymers of the wheat straw biomass will decompose via devolatilization, depolymerization, carbonization, deoxygenation, to various degrees in four important phases, depending on degradation temperature, biomass composition, and other factors [40]. Hemicellulose is the polymer that degrades to the highest extent relative to lignin and cellulose which decompose usually at higher temperatures of >250 °C [40].

Results from FTIR, SEM, TGA, and Py-GC-MS all suggest that the torrefaction temperatures of 280 °C and 300 °C were sufficient to alter physical properties of wheat straw. The representative SEM images of raw and 280 °C-torrefied wheat-straw under the same 45× magnification are depicted in Figure 1. Evidently, the raw biomass sample contains very strong, bulky xylem tissues. The 45× enlargement shows that the biomass began to lose its bound fibrous structure and highly irregularly shaped particles with similar diameters formed upon torrefaction. Structural breakage like cracks and fissures, as well as char micropores and sublayers, became more apparent in the particles during devolatilization, and this increased surface area. The improved fluidization of torrefied particles for subsequent thermochemical processes such as pyrolysis, gasification, or combustion is generally associated with this and an absence of little thread or needle-like fibers at the edge of milled torrefied particles, as confirmed by SEM observations. The raw wheat-straw high-heating-rate chars consist mainly of large, distorted fibrous particles that retain the basic shape of the original biomass particles. Small particles show significantly more loss of initial biomass structures during charring than large particles. The high-heating-rate chars' explosive porous structure is generally assumed to be responsible for this higher reactivity. The images clearly reveal that torrefaction significantly changed the particle microstructure by increasing cellulose porosity and surface area.

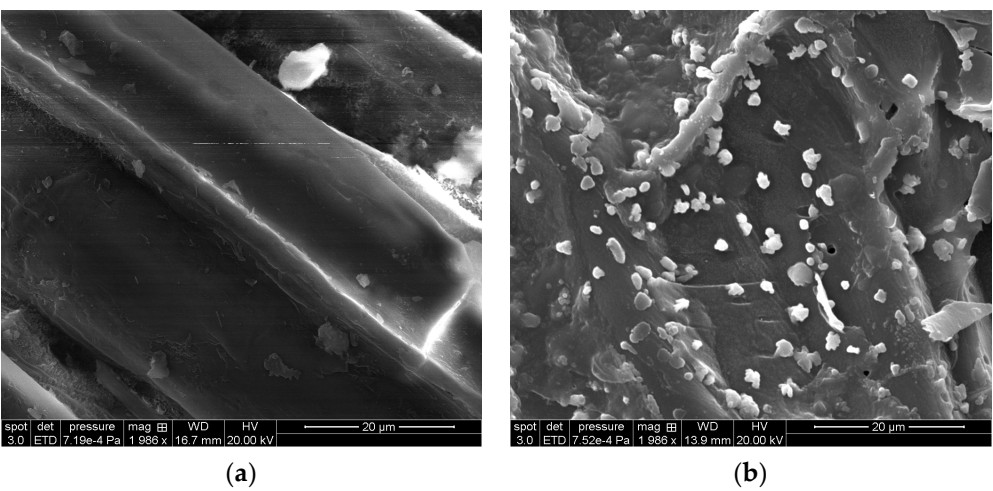

**Figure 1.** SEM imagery of 42–60 mesh wheat-straw particles that were (**a**) not torrefied and (**b**) torrefied at 280 °C.

Fourier transform infrared (FTIR) spectra of the solid raw and 300 °C-torrefied wheat straw also confirmed changes in the chemical structure (Figure 2). FTIR Spectroscopy employs infrared light and is used to detect and identify structural groups and components belonging to molecules of hemicellulose, cellulose and lignin in the torrefied biomass [40]. The infrared light has wavenumber range of 12,800~10 cm$^{-1}$ [40]. The absorption radiation of most organic compounds and inorganic ions is within the 4000–400 cm$^{-1}$ range [40]. Functional groups of interest were those in the regions where most of the transformation was observed such as in the O-H, C-O, C-C, C-H and C-O-C groups. In general, the 300 °C-torrefied wheat straw had its functional group vibrations shifted towards lower wave numbers with changes in intensity. The 1700–1740 cm$^{-1}$ corresponded to carboxylic

acid groups. A shift in the bands at about 1740–1710 cm$^{-1}$ was observed, and these were related to the stretching vibrations of the C-O groups [41]. This vibration was largely due to the degradation of carboxylic acids in hemicelluloses, which include xyloglucan, arabinoglucuronoxylan and galactoglucomannan [15] and can comprise up to 32% of wheat-straw cell wall components. Torrefaction at 300 °C eliminated this signal by decreasing the amount of carboxylic acids groups and generating new products, which appeared at the lower 1700 cm$^{-1}$ wave number. Torrefaction at 300 °C also caused the C-C stretch to move to a lower wave number and increase in intensity. Hemicellulose degradation by torrefaction at 300 °C therefore likely resulted in an increase in unsaturation as more non-polar and unsaturated compounds were generated. Torrefaction at 300 °C also caused a decrease in intensity of bands in the 1250–1220 cm$^{-1}$ region corresponding to the C-O-C vibrations in cellulose. Torrefaction at 300 °C caused a decrease in intensity of bands corresponding to lignin. For instance, the vibrations at 1269 cm$^{-1}$ potentially resulted from the aromatic C-O stretching of methoxyl and phenyl propane units, and those at 1516 cm$^{-1}$ and 1508 cm$^{-1}$ potentially resulted from the C-C aromatic ring vibrations. Torrefaction at 300 °C also caused an increase in aliphatic content and in intensity of bands at 800–900 cm$^{-1}$ and 1600 cm$^{-1}$ that corresponded to a C=C bond common in aromatics. Torrefaction at 300 °C caused a decrease in intensity for bands between 1030 and 1100 cm$^{-1}$ that corresponded to a decrease in C-O bonds, and volatiles and oxygen content. Torrefaction at 300 °C also caused a decrease in intensity in the 3060–3100 cm$^{-1}$ band (data not shown) corresponding to the O-H bonds as H$_2$O was removed to make the wheat-straw more hydrophobic and less susceptible to microbial spoilage during storage.

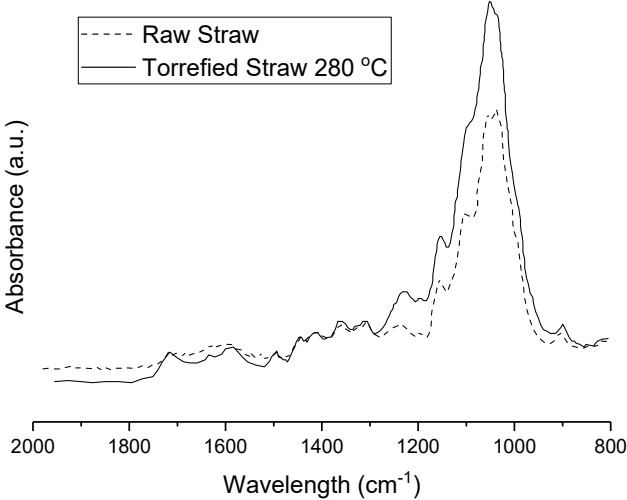

**Figure 2.** FTIR spectra for raw and 280 °C-torrefied wheat-straw particles.

Based on TGA data, approximately 31–32% of the biomass remained as fixed carbon or biochar for both raw and 280 °C-torrefied wheat straw, while the remaining 70% was lost as volatilized compounds that began vaporizing at around 150 °C. Biochemical composition analysis has shown wheat straw to be composed of 18.8% lignin, 18.2% hemicellulose, 52.4% α-cellulose, and 3.7% ash [42]. Hemicelluloses, celluloses, and lignin are known to decompose at 225–300 °C, 305–375 °C, and 250–300 °C [43]. In torrefaction, hemicellulose is considered to be the most reactive constituent and degrades via the removal and decomposition of various saccharides and branches in the hemicellulose. Cellulose, which is composed of a polymer of glucose with no branches, is thermally more stable and thermally decomposes at higher temperatures. Lignin, to a limited extent, will decompose to three products: a char solid, tars and volatiles/gases, the latter two contributing to the significant loss of mass upon torrefaction at higher temperatures.

The plots of conversion vs. either the log (exponential factor) and activation energies for the Friedman method are depicted in Figure 3. Activation energy is a barrier to energy

transfer between reacting molecules that must be overcome and is evidently a function of conversion, likely because of the different primary reactions occurred during thermogravimetric analysis. The Friedman-method average activation energy and exponential factor for all 0.05-incremented conversions between 0.05 and 0.90 for the raw wheat-straw was estimated as 199,398 J mole$^{-1}$ and $1.12 \times 10^{18}$ min$^{-1}$, respectively. Those for the 280 °C-torrefied wheat-straw for conversions between 0.05 and 0.65 were estimated to be 233,406 J mol$^{-1}$ and $2.84 \times 10^{28}$ min$^{-1}$, respectively. Knowledge of the kinetics of biomass thermochemical reactions during torrefaction is very important to estimate the yield of products, understand reaction mechanisms, and enhance selectivity, and design more efficient thermochemical reactors.

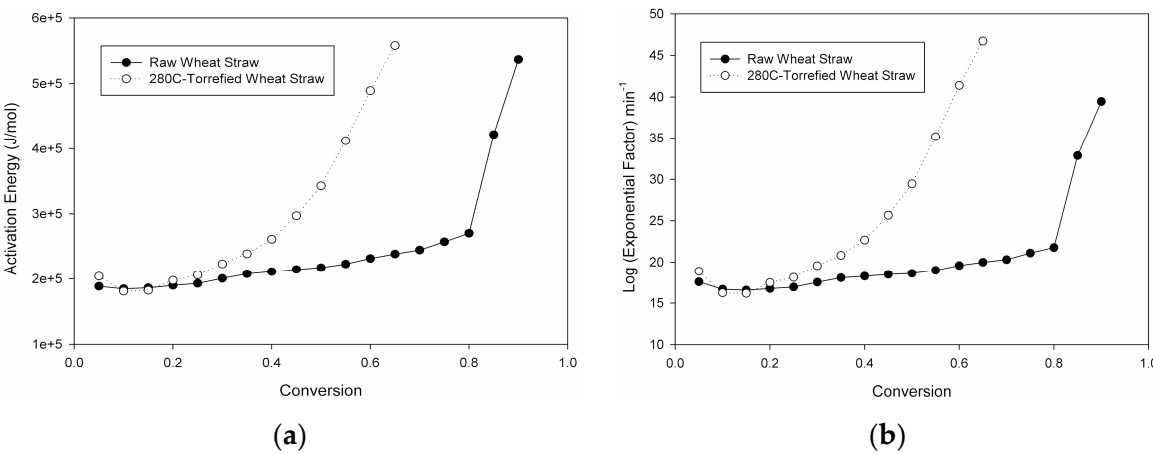

**Figure 3.** Rate exponential factor (**a**) and activation energy (**b**) as function of conversion for raw and 280 °C-torrefied 40–60 mesh wheat-straw particles, as determined by the Friedman Method.

Pyrolysis–gas chromatography–mass spectrometry (Py-GC–MS) is a technique which thermally decomposes a sample's large high-molecular weight molecules (at >600 °C in the absence of $O_2$) into smaller and more volatile low-molecular weight fragments that are then separated by gas chromatography in an inert atmosphere or a vacuum depending on their volatility. Higher volatile particles travel faster through the column than lower volatile particles. The volatile molecules are then ionized using an electric charge and then sent to an electromagnetic field that filters the ions based on their specific mass (mass divided by charge number) expressed in units of m/z, which an ion detector will count. A mass spectrum correspondingly depicts the ion abundance that is counted by a detector via measurement of the current of electrons generated when the ions strike the detector for each m/z as a function of retention time. Spectra from Py-GC MS of raw and 300 °C-torrefied wheat straw revealed a marked change in relative peak areas for various compounds (Figure 4). These primary peaks corresponded to various compounds whose retention times are presented in Table 1.

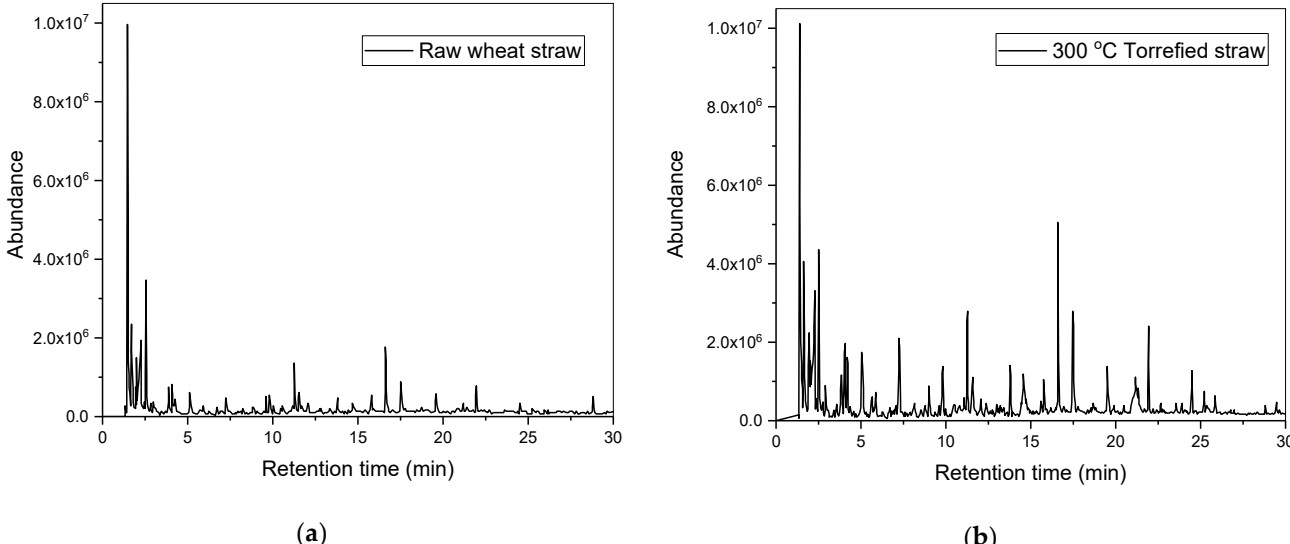

**Figure 4.** Py-GC MS spectra for raw (**a**) and 300 °C-torrefied 42–60 mesh (**b**) wheat-straw particles.

**Table 1.** Properties for compounds detected from py-GC MS.

| Detected Compound | Formula | Retention Time (min) | Molecular Weight (g mole$^{-1}$) |
|---|---|---|---|
| Carbon Dioxide | $CO_2$ | 1.55 | 44 |
| Acetaldehyde | $C_2H_4O$ | 1.872 | 44 |
| 1-propen-2-ol acetate | | 1.761 | |
| Acetic acid | $C_2H_4O_2$ | 2.417 | 60 |
| 2,3-Butanedione | | 2.085 | |
| 2-Propanone, 1-hydroxy- | $C_3H_6O_2$ | 2.65 | 74 |
| 1,2-ethanediol, monoacetate | | 3.961 | |
| Butanediol | | 4.177 | |
| Propanoic acid, 2-oxo-, methyl ester | $C_4H_6O_3$ | 4.324 | 102 |
| Furfural | $C_5H_4O_2$ | 5.135 | 96 |
| 2-Furanmethanol | $C_5H_6O_2$ | 5.734 | 98 |
| 2-Propanone, 1-(acetyloxy)- | $C_5H_8O_3$ | 5.924 | 116 |
| 1, 2-Cyclopentanedione, 3-methyl- | | 9.859 | 112 |
| Phenol, 2-methoxy- | $C_7H_8O_2$ | 11.291 | 124 |
| Pentanal/Cyclopropyl cabinol | | 11.619 | |
| Phenol, 2-methoxy-4-methyl- | $C_8H_{10}O_2$ | 13.815 | 138 |
| Benzofuran, 2,3-dihydro- | $C_8H_8O$ | 14.63 | 120 |
| Phenol, 4-ethyl-2-methoxy- | $C_9H_{12}O_2$ | 15.82 | 152 |
| 2-Methoxy-4-vinylphenol | $C_9H_{10}O_2$ | 16.632 | 150 |
| Phenol, 2,6-dimethoxy- | $C_8H_{10}O_3$ | 17.507 | 154 |
| Phenol, 2-methoxy-4-(1-propenyl)- | $C_{10}H_{12}O_2$ | 19.569 | 164 |
| 5-tert-Butylpyrogallol | | 21.148 | |
| 2-propanone, 1-(4-hydroxy-3-methanyphenyl-) | | 21.342 | |
| 3,5-dimethoxyacetophenome | | 21.929 | |
| Phenol, 2,6-dimethoxy-4-(2-propenyl)- | $C_{11}H_{14}O_3$ | 24.496 | 194 |
| ethanone, 1-(4-hydroxy-3,5 dimethoxyphenyl-) | | 25.203 | |
| desaspidinol | | 25.855 | |
| 3,5-dimethoxy-4-hydroxycinammaldehyde | | 29.457 | |
| Estra-1,3,5(10)-trien-17á-ol | $C_{18}H_{24}O$ | 28.793 | 256 |

### 3.2. Analysis of Wheat-Straw Enzymatic Hydrolysate

The final $C_5$ and $C_6$ sugar yields from 280 °C-torrefied and raw wheat straw after $O_3$/SAA pre-treatment and enzymatic hydrolysis were very similar and differed by 12.8% and 16.0%, respectively (Table 2). This suggests that the increased hydrophobicity of the torrefied biomass did not adversely impact a pre-treatment requiring swelling of

the biomass and that enough hemicellulose remained after the torrefaction process to provide $C_5$ sugars. Instead of the conventional dilute sulfuric acid pre-treatment [44], the novel pre-treatment method involving $O_3$ and soaking aqueous $NH_4OH$ followed by enzymatic hydrolysis described herein provided the advantages of more economical, milder operating conditions, recyclable $(NH_4)_2(SO_4)$ for algal cultivation, and a lower yield of growth-inhibiting compounds [10]. The acetate-buffered enzymatic hydrolysate contained 27.49 mg $L^{-1}$ glucose, 0.69 mg $L^{-1}$ arabinose, and 10.06 mg $L^{-1}$ xylose/mannose. $^1H$ NMR spectra of enzymatic wheat-straw hydrolysate depicted sugars in the 3–4 ppm region and not small molecules or growth inhibitors in the 6–10 ppm region (Figure 5). This study represents the first reported use of torrefaction to prepare biomass for subsequent biochemical pre-treatment and hydrolytic enzymatic processes.

**Table 2.** Comparative reducible and yielded $C_5$ and $C_6$ sugar content in both torrefied and non-torrefied wheat-straw using acetate or citrate buffer.

| | Pre-Treatment | | | |
|---|---|---|---|---|
| | $O_3$ | 280 °C Torrefaction + $O_3$ | $O_3$ + $NH_4$ | 280 °C Torrefaction + $O_3$ + $NH_4$ |
| **Carbohydrate** | **Concentration (ppm)** | | | |
| Arabinose | 0.39 | 0.33 | 0.73 | 0.59 |
| Galactose | 0.06 | 0.04 | N/A | N/A |
| Glucose | 11.25 | 11.39 | 25.88 | 22.57 |
| Xylose/Mannose | 5.16 | 4.72 | 10.30 | 8.64 |

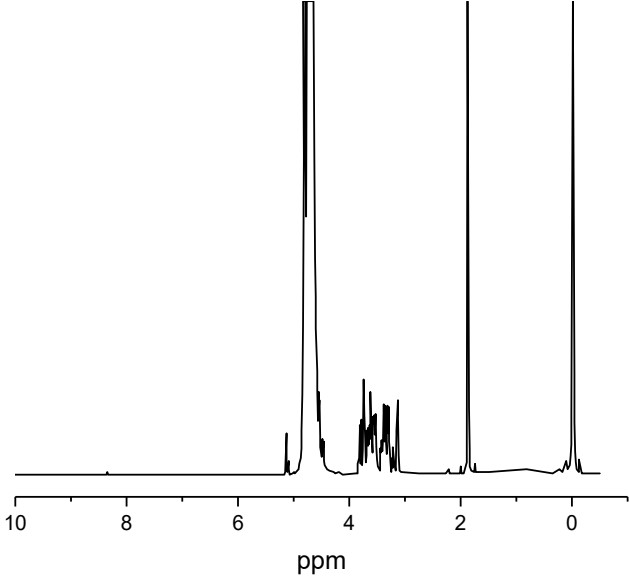

**Figure 5.** $^1H$ NMR spectra of enzymatic wheat-straw hydrolysate depicting sugars in the 3–4 ppm region and not small molecules in the 6–10 ppm region.

### 3.3. Analysis of Liquid Condensate from Wheat-Straw Torrefaction

Results from GC analysis of the initial torrefaction liquid condensate show that carboxylic acid content increased as torrefaction temperature increased from 260 °C to 300 °C (Table 3). This suggests that more hemicellulose was degraded via primary thermochemical reactions as temperature increased. Importantly, the greatest amount of acetic acid of 72,998 ppm as a cheap and sustainable source of organic carbon and/or pH adjuster for microalgal cultivation was obtained at the highest temperature of 300 °C. Although the permanent gases formed during this torrefaction were not analyzed, the composition of these gases mainly includes CO, $CO_2$, trace amounts of $CH_4$, and low molecular weight hydrocarbons (CxHy) combined with unreacted $H_2$, $O_2$ and $N_2$. The $CO_2$ and CO contents

normally range ~75–80% and 10–12% of all gaseous products [40]. CO represents the majority of the permanent gas heating value during torrefaction [40] and can be used as fuel to achieve the high 300 °C temperature corresponding to maximal acetic acid yield.

**Table 3.** Comparative carboxylic acid and ethanol content in liquid condensate arising from torrefied wheat-straw at 260 °C, 280 °C, and 300 °C × 20 min in an Auger reactor.

| | Torrefaction Temperature | | |
| --- | --- | --- | --- |
| | **260 °C** | **280 °C** | **300 °C** |
| **Compound** | Concentration (ppm) | | |
| Ethanol | 1097 | 4879 | 8751 |
| Acetic Acid | 2533 | 47,944 | 72,998 |
| Propionic Acid | 367 | 2996 | 7926 |
| Isobutyric Acid | 151 | 518 | 1198 |
| Butyric Acid | 0 | 634 | 1292 |
| Isovaleric Acid | 364 | 2334 | 4248 |
| Valeric Acid | 91 | 754 | 1931 |
| Isocaproic Acid | 80 | 209 | 632 |
| Caproic Acid | 555 | 1836 | 4013 |
| Heptanoic Acid | 193 | 604 | 1270 |

The inability of ALP2 to grow on $100\times$ diluted condensate made it necessary to remove such inhibitory compounds and coloration with a neutralization, purification, and dilution scheme at the risk of simultaneously lowering the yield of available carboxylic acids. For instance, only 52.17% of the acetic acid remained after de-colorization. Therefore, future work should be conducted to replace or optimize this de-colorization step. In the approach used herein, the $O_3$ used for biochemical pre-treatment was also used here to treat activated carbon and increase the number of its acidic surface residues to repel desirable acetate ions away from carbon. [1]H NMR analysis of the 300 °C-torrefaction condensate before and after activated-carbon adsorption revealed a thorough removal of small molecules such as potentially inhibitory furfurals, furans, HMF, and acetol in the leftmost corresponding ppm region (Figure 6). [1]H NMR analysis of vaporized products that were collected during vacuum distillation revealed the presence of acetol (data not shown). The solid char particle product from an integrated wheat-straw slow-pyrolysis process can not only by used by local farms to improve soil and $CO_2$ sequestration, but also, once activated with KCl pre-treatment, used for this activated carbon adsorption. The GC-based carboxylic acid content of the purified, de-toxified, filtered 300 °C-torrefaction liquid condensate is presented in Table 4. This was equivalent to 62.25 g L$^{-1}$ potassium acetate and was diluted $100\times$ for algal growth experiments.

**Table 4.** Carboxylic acid content in purified, de-toxified liquid condensate arising from torrefaction of wheat-straw at 300 °C.

| Compound | Concentration (ppm) |
| --- | --- |
| Acetic Acid | 38,082 |
| Propionic Acid | 2689 |
| Isobutyric Acid | 109 |
| Butyric Acid | 99 |
| IsovalericAcid | 244 |
| Caproic Acid | 45 |

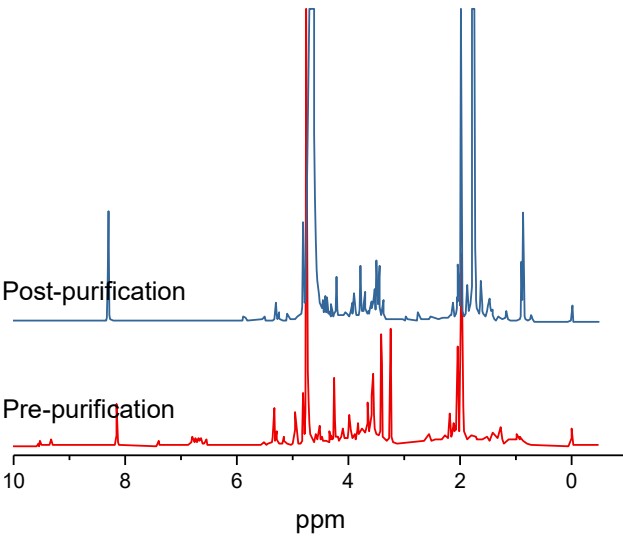

**Figure 6.** $^1$H NMR spectra of wheat-straw torrefaction condensate before (**bottom**) and after (**top**) purification and de-toxification.

*3.4. Analysis of Effluents from Anaerobically Digested Food Waste and Dairy Manure*

　　The physical properties of both the anaerobically digested flush dairy manure and food waste effluents before and after autoclaving are presented in Table 5. The majority of nitrogen in anaerobically digested waste effluents is in the form of ammonium and nitrate salts and magnesium or calcium phosphate salts. The majority of N-ammonia did not volatilize during high temperature autoclaving. It is possible that high ionic strength [45] from additionally supplemented $NaHCO_3$ salts and alkalinity used to supply inorganic carbon and limit contamination, as well as the microalgal sorbent properties [46], can further help reduce ammonia volatilization during the proposed outdoor phototrophic second-stage cultivation at high outdoor summer temperatures and high pH 10. An initial and final effluent total nitrogen volatilization rate of 16.15 mg L$^{-1}$ day$^{-1}$ and 3.66 mg L$^{-1}$ day$^{-1}$ were measured for a foam-capped 0.25 L flask containing 16.8 g L$^{-1}$ $NaHCO_3$ and 5% (*v/v*) of anaerobically digested food waste effluent in this study based on a plot of media nitrogen as a function of time (Figure 7). For a similar culture flask, a relatively low overall equivalent gas–liquid $O_2$-mass transfer coefficient of 24 h$^{-1}$ was previously estimated for similarly foam-capped 0.25 L shaker flasks [47].

**Table 5.** Measured properties of anaerobically digested effluents.

|  | **Food Waste** | **Dairy Manure** |
|---|:---:|:---:|
| **Effluent Properties (g L$^{-1}$)** | | |
| **Pre-Autoclave** | | |
| Ammonia-N | 0.89 | 0.63 |
| Total-N | 1.07 | 0.71 |
| Alkalinity | 7.04 | 2.35 |
| pH | 7.79 | 7.32 |
| $PO_4^{3-}$ | 0.08 | 0.42 |
| COD | 6.64 | 15.00 |
| **Post-Autoclave** | | |
| Ammonia-N | 0.66 | 0.24 |
| Alkalinity | 5.90 | 2.29 |
| pH | 9.26 | 9.23 |

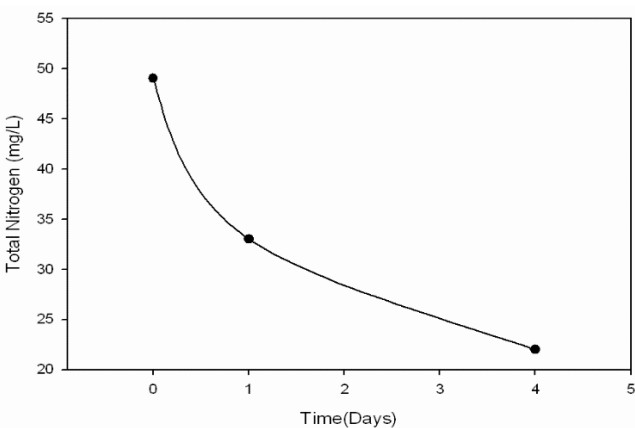

**Figure 7.** Total nitrogen remaining in a flask containing BG-11$_0$ medium supplemented with 16.8 g L$^{-1}$ NaHCO$_3$ and 2.5% (*v/v*) anaerobically digested food waste effluent, initially adjusted to pH 9.

The ammonia volatilization rate in a microalgal open-pond can be predicted via a convective mass transfer model dependent on ambient temperature, airflow over the surface, pH, ionic strength, and the $NH_3$ concentration gradient between the ammonium ($NH_4{}^+$) solution and the ambient air estimated as per Equation (10).

$$J_{NH3} = -K_m \left( [NH_3]_{gas} - [NH_3]_{ambient} \right) \tag{10}$$

where $J_{NH3}$ is the $NH_3$ volatilization flux (g $NH_3-N$ m$^{-2}$ s$^{-1}$), $K_m$ is the convective mass transfer coefficient (m s$^{-1}$), $[NH_3]_{gas}$ is the concentration of gaseous $NH_3$ in equilibrium with dissolved $NH_3$ in solution (g $NH_3-N$ m$^{-3}$ air), and $[NH_3]_{ambient}$ is the concentration of $NH_3$ in ambient air (g $NH_3-N$ m$^{-3}$ air), which is normally very small and assumed to be negligible. The concentration of gaseous $NH_3$ at equilibrium with the dissolved $NH_3$ in solution is determined using Henry's law relating to the concentration of dissolved $NH_3$ in the BG-11 medium supplemented with food waste effluent to an equilibrium concentration of $NH_3$ in the air space immediately above the liquid surface following Equation (11).

$$K_H = \frac{[NH_3]_{gas}}{[NH_3]_{solution}} = \left( \frac{0.2138}{T} \right) 10^{6.123 - \frac{1825}{T}} \tag{11}$$

where $K_H$ is the Henry's law constant, (g $NH_3-N$ m$^{-3}$ air)/(g $NH_3-N$ m$^{-3}$ solution) and $T$ is absolute temperature of BG-11 media (K). Ammoniacal N in the BG-11 media consists of both $NH_3-N$ and $NH_4{}^+-N$. The fraction of total ammoniacal $N$ present as $NH_3$ is estimated using equilibrium thermodynamic principles as per Equation (12).

$$f_{NH3} = \frac{[NH_3]_{solution}}{[NH_3]_{solution} + [NH_4^+]_{solution}} = \frac{1}{1 + \frac{10^{-pH}}{K_a'}} \tag{12}$$

where $K_a{}'$ is the dissociation constant corrected for ion effects in the high bicarbonate salt-supplemented BG-11 medium, $pH = -\log[H^+] = 10$, and $f_{NH3}$ is the $NH_3-N$ fraction of the total ammoniacal $N$ of the BG-11 media. The corrected disassociation constant is obtained from Equation (13).

$$K_a' = K_a \frac{\gamma_{NH_4^+}}{\gamma_{NH_{3solution}} \gamma_{H^+}} \tag{13}$$

where $\gamma$ represents activity coefficients, and $K_a$ is uncorrected disassociation constant. $K_a$ is obtained following Equation (14) [48].

$$K_a = 100^{0.05 - \frac{2788}{T}} \tag{14}$$

and the activity coefficient for ammonium is obtained following Equation (15) [49].

$$\log \gamma_{NH_4^+} = -0.5a^2 \left( \frac{\sqrt{I}}{1 + \sqrt{I}} - 0.3I \right) \tag{15}$$

where *a* is the charge $1^+$ of the ion $NH_4^+$ and *I* is the ionic strength. For algae and duckweed-based stabilization ponds treating domestic wastewater [50], the mass transfer coefficient $K_m$ is obtained following Equation (16).

$$K_m = \left( \frac{0.0566}{d} \right) e^{0.13(T-20)} \tag{16}$$

where *d* is the depth of the outdoor microalgal open-pond PBR (m). Therefore, the open-pond depth should be shallow enough to allow light penetration but high enough to prevent ammonia volatilization when using effluents. Ammonium ($NH_4^+$) removal efficiencies of $64.5 \pm 2.8\%$ and $51.2 \pm 1.9\%$ were achieved in shallow and deep ponds, respectively [50]. Alternatively, the following correlation (Equation (17)) from dimensional analysis can be used to determined mass transfer coefficient:

$$Sh = b\mathrm{Re}^n Sc^m \tag{17}$$

where *Sh* = Dimensionless Sherwood number, Re = Dimensionless Reynold's number, *Sc* = Dimensionless Schmidt number, and *b*, *n*, and *m* are dimensionless coefficients [51].

To draw more definitive conclusions and enable design of appropriate ammonia-containing effluent fed-batch algal cultivation strategies, future studies should be conducted to reconcile predicted and measured ammonia volatilization rates from open-pond systems.

### 3.5. Microalgal Growth Experiments

Comparative heterotrophic growth of ALP2 microalgae on 10 g $L^{-1}$ glucose or acetate-buffered hydrolysate corresponding to the same glucose concentration is shown (Figure 8). The hydrolysate culture resulted in a final *DCW* of g $L^{-1}$ and % higher *DCW* than that achieved on glucose alone. Similarly superior growth on hydrolysate was previously witnessed when *Chlorella pyrenoidosa* was heterotrophically cultivated using hydrolysate from lignocellulosic rice straw waste pretreated using trifluoroacetate at 95 °C and ethanol at 210 °C [6].

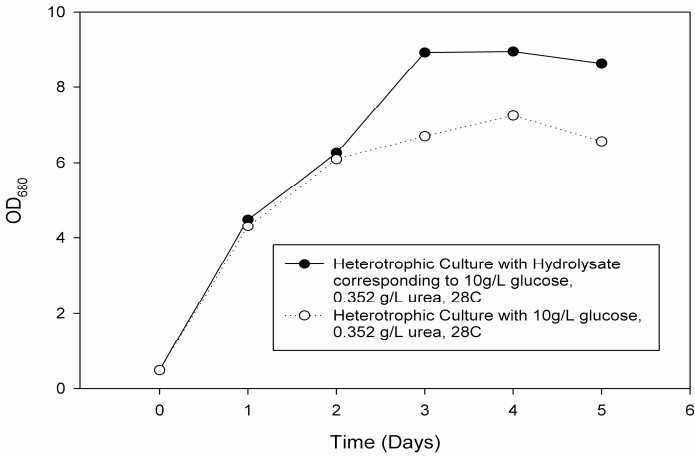

**Figure 8.** Comparative heterotrophic growth of ALP2 on hydrolysate and 10 g $L^{-1}$ glucose.

Food waste effluent 5% (*v*/*v*) dilutions at both initial pH 8 and 9 resulted in higher final absorbance readings for ALP2 cultures compared to those involving flush dairy manure

effluent (data not shown). Flow cytometry also demonstrated segregation of green ALP2 microalgae from light-blocking particulate matter of the dairy manure effluent that was absent in food waste effluent (Figure 9).

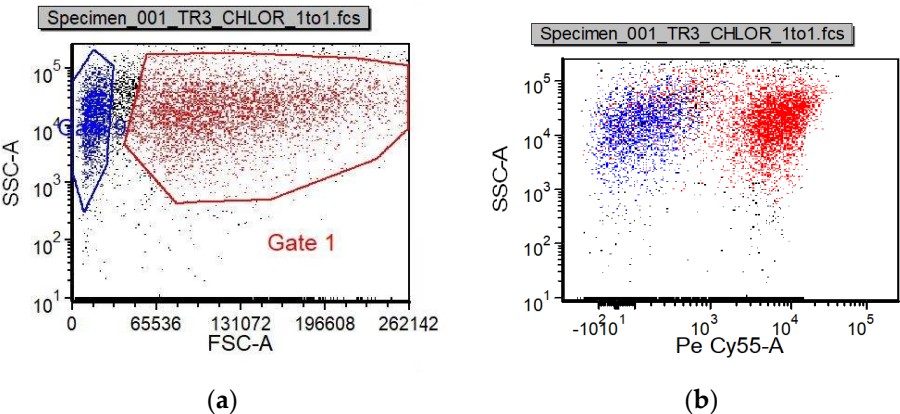

**Figure 9.** Flow cytometric forward scatter vs. side scatter plot (**a**) and PE Cy 5.5 Fluorescence vs. Side Scatter Plot (**b**) of gated anaerobically digested manure particles and ALP2 microalgal populations.

Food waste effluent was further evaluated instead because of its greater optical clarity and superior growth performance. The combined effect of $NaHCO_3$ concentration, food waste effluent dilution, and periodic pH adjustment with acid on ALP2 growth and lipid accumulation is presented in Table 6. Consistent with induction of nitrogen-depletion stress, cultures with minimal 5% (*v/v*) effluent levels exhibited the lowest final *DCW* but generally resulted in the highest lipid content compared to 10%, 15%, and 20% (*v/v*). High salt and $NaHCO_3$ alkaline levels prolonged an acclamatory lag-phase but did not limit the ability of ALP2 to reach high biomass when pH was not daily adjusted. ALP2 initially grew the fastest and reached an extremely high pH and stationary phase in cultures containing the least amount of $NaHCO_3$ where buffering was minimal for all effluent levels. For instance, a pH of 11.84 was reached at 8.4 g $L^{-1}$ $NaHCO_3$ and 5% (*v/v*) effluent. Such a stationary phase may be attributed to ammonia inhibition occurring at high pH. Daily pH 10 adjustment with HCl acid for the 8.4 g $L^{-1}$ $NaHCO_3$ and 20% effluent condition resulted in a significant 100.89% increase in the $OD_{680}$ after 120 h, higher total fat content, and a more saturated fatty acid profile, compared to no pH adjustment (data not shown). For similar cultures, the normalized BODIPY fluorescence, indicative of relative neutral lipid content, was 4.97, 3.04, 15.54, and 8.43 for 5% AD effluent + 8.4 g $L^{-1}$ $NaHCO_3$, 20% AD effluent + 8.4 g $L^{-1}$ $NaHCO_3$, 5% AD effluent + 33.6 g $L^{-1}$ $NaHCO_3$, 20% AD effluent + 33.6 g $L^{-1}$ $NaHCO_3$, respectively (Figure 10).

**Table 6.** Comparative ALP2 lipid content at various NaHCO₃ and ADFW levels.

| NaHCO₃ (g L⁻¹) | ADFW (%) | pH Unadjusted | | | pH Adjusted Daily to 10 | | |
|---|---|---|---|---|---|---|---|
| | | *DCW* (g L⁻¹) | Lipid (%) | Lipid (g L⁻¹) | *DCW* (g L⁻¹) | Lipid (%) | Lipid (g L⁻¹) |
| 8.4 | 5 | 0.971 | 24.11 | 0.234 | 1.140 | 34.94 | 0.398 |
| | 10 | 0.905 | 17.28 | 0.156 | 1.389 | 28.50 | 0.396 |
| | 15 | 0.932 | 14.85 | 0.138 | 1.604 | 26.65 | 0.427 |
| | 20 | 0.927 | 11.46 | 0.106 | 1.951 | 25.14 | 0.491 |
| 16.8 | 5 | 1.128 | 33.06 | 0.373 | 1.095 | 38.87 | 0.426 |
| | 10 | 1.173 | 23.44 | 0.275 | 1.421 | 29.88 | 0.425 |
| | 15 | 1.173 | 19.53 | 0.229 | 1.619 | 26.11 | 0.423 |
| | 20 | 1.125 | 17.04 | 0.192 | 1.699 | 25.90 | 0.440 |
| 25.2 | 5 | 0.883 | 34.91 | 0.308 | 0.785 | 30.69 | 0.241 |
| | 10 | 1.058 | 29.71 | 0.314 | 1.094 | 27.89 | 0.305 |
| | 15 | 1.287 | 21.94 | 0.282 | 1.272 | 23.55 | 0.300 |
| | 20 | 1.211 | 14.86 | 0.180 | 1.213 | 20.41 | 0.248 |
| 33.6 | 5 | 0.694 | 25.53 | 0.177 | 0.591 | 15.70 | 0.093 |
| | 10 | 1.109 | 31.28 | 0.347 | 0.938 | 19.76 | 0.185 |
| | 15 | 1.286 | 19.89 | 0.256 | 1.066 | 18.59 | 0.198 |
| | 20 | 1.223 | 15.13 | 0.185 | 0.716 | 19.37 | 0.139 |

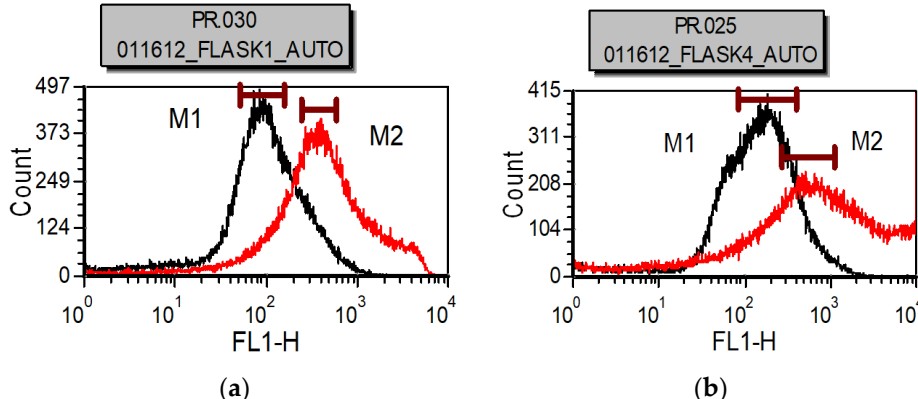

**Figure 10.** Flow cytometric FL1 histogram plots for BODIPY-stained ALP2 cultures on 5% (*v/v*) (**a**) and 20% (*v/v*) (**b**) anaerobically digested food waste effluent.

At 8.4 g L⁻¹ NaHCO3, the 5% vs. 22% AD effluent lipid content ratio was 1.64, based on flow cytometry and BODIPY, compared to 1.39 using FAME GC-FID. The total fat content and fatty acid profile of the ALP2 culture at 8.4 g L⁻¹ NaHCO₃ and 20% effluent with acid addition is shown in Table 7. The four prominent fatty acids were palmitic (C16:0), oleic (C18:1n9), linoleic (C18:2n6), and linolenic (C18:3n3). Interestingly, oleic, linoleic, and linolenic composed 50% of the total fatty acids. To the best of our knowledge, the only oil with a higher concentration of omega-3 linolenic fatty acid is flaxseed oil. With such a favorable omega-6/omega-3 ratio, the results on the residues of this microalgae after and neutral TAG lipid extraction suggest these residues could be used as substitutes of omega-6/omega-3. Compared to such lipids derived from microalgal mixotrophic cultivation that are subsequently extracted and trans-esterified to biodiesel, a bio-oil intermediary product from high temperature fast-pyrolysis or hydrothermal liquefaction will subsequently require a series of more complex and expensive hydrotreatment, fractional distillation, and other bio-refining steps [7].

**Table 7.** ALP2 fatty acid profile at 8.4 g L$^{-1}$ NaHCO$_3$ and 20% (*v/v*) ADFW levels.

| Effluent Component | Readings |
|---|---|
| **Pre-Autoclave** | |
| Alkalinity | 7.04 g L$^{-1}$ |
| Ammonia-N | 0.89 g L$^{-1}$ |
| Total-N | 1.07 g L$^{-1}$ |
| PO$_4$$^{3-}$ | 0.08 g L$^{-1}$ |
| COD | 6.64 g L$^{-1}$ |
| pH | 7.79 |
| **Post-Autoclave** | |
| Alkalinity | 5.90 g L$^{-1}$ |
| Ammonia-N | 0.66 g L$^{-1}$ |
| pH | 9.26 |

The two-stage cultivation strategy was finally demonstrated using these alternative sources of organic carbon, nitrogen, and phosphorous. ALP2 rapidly grew mixotrophically on both glucose-containing hydrolysate and acetate-containing condensate and achieved a cell productivity of $7.25 \times 10^7$ cells mL$^{-1}$ day$^{-1}$ and $2.25 \times 10^7$ cells mL$^{-1}$ day$^{-1}$, respectively (Figure 11). Evidently, ALP2 was able to grow on this condensate and hydrolysate without acclimatory lag phase, suggesting that growth inhibitors were successfully removed.

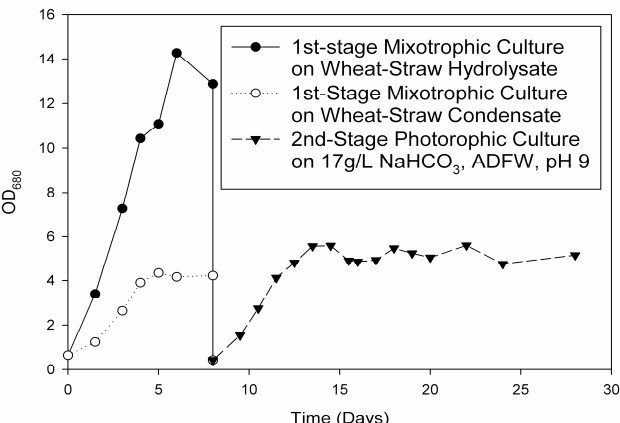

**Figure 11.** $OD_{680}$ as a function of time during two-stage cultivation of ALP2 using integrated waste.

It appears that there are no reports on the feasibility of using the carboxylic acid content of torrefaction liquid condensate in other potential integratable biorefinery applications such as microalgal cultivation. The acidic liquid condensate could alternately be used as organic carbon source for acidogens and methanogens in an integrated AD process to provide biogas for on-site energy or as previously described pH adjustment for second-stage phototrophic outdoor open-ponds. Although ALP2 grew faster and yielded higher biomass and cell density mixotrophically than heterotrophically, a biorefinery mixotrophic first-stage process for two-stage biofuel production would today likely suffer from limited availability of validated scaled-up studies or high capital costs of photobioreactors (PBRs) [52,53]. These would require, for example, sterilizable, durable, and non-discoloring enclosed polycarbonate body, flashing red and blue LEDs [54] and fluorescent bulbs, and acrylic light-guides needed for greater light penetration. These high costs of PBRs in the future could be offset by additional high revenues from cGMP-compliant and highly controlled manufacturing of FDA-regulated, high-value neutraceuticals such as lutein or astaxanthin [55]. For instance, the microalgae *Haematococcus pluvialis* is the highest producer of the high-value astaxanthin most optimally grown mixotrophically using 30 mM acetate in a so-called "green-stage" [56].

The two mixotrophic cultures were then used to inoculate at high cell density a second phototrophic stage featuring fed-batch added food waste effluent. The variation in

nitrogen levels arising from the fed-batch strategy in the phototrophic second-stage is depicted in Figure 12. In other studies involving Spirulina platensis, batch-fed supplementation of ammonium sulfate was shown to limit this toxicity as well as volatilization [38]. Based on $^1$H NMR spectra, the second stage culture achieved a volumetric lipid productivity of 0.045 g L$^{-1}$ day$^{-1}$ (Figure 13). This is comparable to the reported volumetric productivity of a fast-growing, oleaginous photosynthetic *Chlorella vulgaris* algae (29.5 mg L$^{-1}$ day$^{-1}$) [57].

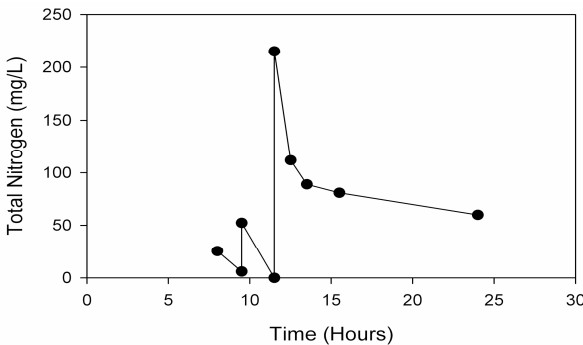

**Figure 12.** Extracellular total N as function of time during fed-batch addition of anaerobically digested food waste effluent for the second phototrophic stage.

ALP2 Food Waste Effluent 17g/L NaHCO3
Jan-29-2013 500 MHz, 128 scans

16.367

1.000

**Figure 13.** $^1$H NMR spectra for intracellular neutral TAG lipid content with integrated peak areas normalized to TMSP standard.

## 4. Conclusions

An improved two-stage microalgal cultivation process for biofuels and co-products that more sustainably used organic carbon, nitrogen, and phosphorous derived from the integrated treatment and conversion of lignocellulosic waste and food waste was here demonstrated in small-scale biorefinery process. Innovations include a novel thermochemical (i.e., torrefaction) and biochemical pretreatment process to generate wheat-straw hydrolysate containing glucose and condensate containing potassium acetate for first-stage mixotrophic cultivation. In addition, a novel fed-batch strategy was employed to supplement optically clear effluent from anaerobic digestion of food waste as N and P supply in a second-stage phototrophic cultivation that featured high $NaHCO_3$, high pH-stat phototrophic conditions previously used to limit contamination, supply inorganic carbon, and improve lipid productivity. The findings suggest that the integration of torrefaction with biochemical pretreatment of lignocellulosic waste and anaerobic digestion of food waste in a biorefinery positively impacts the cultivation of microalgae by cost-effectively supplying organic carbon for a first of two stages.

**Author Contributions:** Conceptualization, P.C.W. and S.C.; methodology, P.C.W.; software, P.C.W. and M.R.P.-S.; validation, P.C.W.; formal analysis, P.C.W.; investigation, P.C.W., M.B., A.G., H.K., W.C.D., G.L.H., W.H., M.G.-P., L.Y., M.R.P.-S.; resources, W.C.D., H.K., G.L.H., W.H., M.G.-P., S.C.; data curation, P.C.W., G.L.H., W.H., W.C.D.; writing—original draft preparation, P.C.W.; writing—review and editing, P.C.W., M.R.P.-S., S.C.; visualization, P.C.W. and M.R.P.-S.; supervision, S.C.; project administration, S.C.; funding acquisition, S.C. All authors have read and agreed to the published version of the manuscript.

**Funding:** This research was funded by Washington State University.

**Institutional Review Board Statement:** Not applicable.

**Informed Consent Statement:** Not applicable.

**Data Availability Statement:** Not applicable.

**Conflicts of Interest:** The authors declare no conflict of interest.

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
