# Peer review of "Biorefinery Processing of Waste to Supply Cost-Effective and Sustainable Inputs for Two-Stage Microalgal Cultivation"

_applsci, doi:10.3390/app12031485_

Round 1
Reviewer 1 Report
The Paper is very interesting. Th results are clear but it would be better to clarify the experimental design and methods.
Author Response
The Paper is very interesting. The results are clear but it would be better to clarify the experimental design and methods.
Answer: Thank you for your comment. We agree that the experimental section needed revision, especially for clarifying part of the methodology. Therefore, we have added references to this section, added details, and explained the meaning of abbreviations. In this section, we have also edited the units for consistency.

Reviewer 2 Report
Please see the attachment.

Author Response
1. A thermochemical and biochemical pre-treatment process to produce wheat straw hydrolysate and a bed-batch to supplement effluent from anaerobic digestion of food waste were proposed and experimentally investigated in this study. The topic of this manuscript is interesting and the manuscript is well structured. The results of this study could provide valuable reference for relevant academician and practicians. There are a few typo or cited-reference errors to be corrected. Minor revision is suggested for this article.
Answer: Thank you. We want to acknowledge the reviewer’s comments on the manuscript. We have conducted a throughout revision of the references and added the missing ones.
2. A few references are missed to cite such as those at lines 68, 166, 175, 177, reference after “method” at lines 198, 217, 236, 243, 264, 276, 488, 698, and 699, etc. In addition, the reference format should be consistent, some reference cited in the format in the last name of authors could be corrected to be in numerical sequence, such as [Ma..] at line 286, [Carvalho …] at line 366, [Rousset …] at line 454, [Van…] at line 456, [Aden…] at line 525, [Chaoui…] at line 614, [Morel…] at line 615, [Zimmo…] at lines 618 and 624, and [Carvalho…] at line 707, etc.
Answer: We have conducted a throughout revision of the references and added the missing ones. We have also added new references as needed.
3. Uncertainties and or error bars of those measured data could be supplemented.
Answer: In the manuscript we have added that all tests were conducted in duplicates only (See Section 2.2), unless otherwise noted. Thus, we are not able to include error bars in our results.
4. Total words could be provided when their abbreviations first appear, such as SEM, ATR, DTG, etc.
Answer: We have provided details on the meaning of all abbreviations as per the reviewer’s request.
5. The meanings of those symbols in the equations such as Eqns. (2) like mn+1 , dt, mn and Eqn. (8) like mt could be explained.
Answer: Details on the meaning of missing symbols in equations have been added.
6. Subtitles could be provided (such as (a)…, (b)….) if two sub-figures (like Figs. 2, 3, 4, 9, 10) or two sub-tables (like Table 4) are presented together.
Answer: Subtitles have been added to subfigures as the reviewer’s suggestion.
7. Important findings of this study should be adequately summarized in the section Conclusion.
Answer: The Conclusion section has been expanded accordingly.
Please see the attachment. Thank you again.

Reviewer 3 Report
The paper titled "Biorefinery Integrating Thermo-Biochemical Pre-treatment and Hydrolysis of Wheat-Straw and Anaerobic Digestion of Food Waste with Two-Stage Microalgal Cultivation" reports interesting data that are useful to consider to publish on Applied Science.
All the sections of the manuscript appear adequate, but not the Reference both when cited in the text, and on the final list. This section need a deep full revision. See details, in the text:
p.2, line 68 in the [ ] no refer.;
p.3, line 124-129 refs,18 and 20 but n. 19 is cited at line 144 !
p.4, line 166 [ ] ; and lines 175-177 [ ] no refs.
p.5, line 217
p.6 line 236 and 243, 264, 276 " "
p. 7, line 286 [Ma et al 2011] Change this ref as Number
p. 10, Fig. 1 (A and B)...I suggest Top and Bottom
p. 22, lines 698-699.No refsrences
p. 23, lines 705 Spirulina platensis in italic; and line 707 (Caralho et al 2004) change with number ref.
Author Response
The paper titled "Biorefinery Integrating Thermo-Biochemical Pre-treatment and Hydrolysis of Wheat-Straw and Anaerobic Digestion of Food Waste with Two-Stage Microalgal Cultivation" reports interesting data that are useful to consider to publish on Applied Science.
All the sections of the manuscript appear adequate, but not the Reference both when cited in the text, and on the final list. This section need a deep full revision. See details, in the text:
p.2, line 68 in the [ ] no refer.;
p.3, line 124-129 refs,18 and 20 but n. 19 is cited at line 144 !
p.4, line 166 [ ] ; and lines 175-177 [ ] no refs.
p.5, line 217
p.6 line 236 and 243, 264, 276 " "
- 7, line 286 [Ma et al 2011] Change this ref as Number
- 10, Fig. 1 (A and B)...I suggest Top and Bottom
- 22, lines 698-699.No refsrences
- 23, lines 705 Spirulina platensis in italic; and line 707 (Caralho et al 2004) change with number ref.
Answer: Thank you very much. We acknowledge that several references were missing in the manuscript submitted. This comment has been raised by the other reviewers too. Consequently, we have revised the whole manuscript and added the missing references. Please see attachment.

Reviewer 4 Report
- The authors focused more on microalgae in the abstract. It looks like a different work from your article title. It is better to rewrite the abstract about the purpose and title of the work.
- Lines 68 and 78, did you remove the citions? … operating conditions []. … after fall harvests [].
- Lines 166, 175 and 177 - why did you add “[]”?
- Please use the uniform symbols of the degree in lines 166,171,172,183,185. And use Celsius or Kelvin in lines 183-185.
- It is better to present a schematic figure for the two-step process.
- Check references on lines 243, 264, 276, and more… Line 366 – Carvalho et al. citation style.
- There was no discussion in the SEM part. And it was similar in many other parts.
- Why didn't you present the FTIR spectrum range between 650 to 4000 (Figure 2) when giving data that is not in the figure in the text (eg 2800-3000 and 700-900)?
- Section 3.3 should be expanded.
- In general, all references should be rechecked as some are missing or out of journal style.
- The discussion part was too weak, it should be revised.
- The aim of the article and the focused results were completely different, the article should be reviewed and re-evaluated.
- The introduction part was too long.
Author Response
- The authors focused more on microalgae in the abstract. It looks like a different work from your article title. It is better to rewrite the abstract about the purpose and title of the work.
Answer: Thank you very much for this and the following insightful feedback which enable improvement of our work. We have re-written the abstract for better clarity and to better reflect the purpose and title of the work for greater consistency. Please see attachment.
- Lines 68 and 78, did you remove the citions? … operating conditions []. … after fall harvests [].
- Lines 166, 175 and 177 - why did you add “[]”?
Answer to points 2 and 3: The brackets are used for references that were missing in the manuscript submitted. We have edited/revised the whole document and fixed this problem.
- Please use the uniform symbols of the degree in lines 166,171,172,183,185. And use Celsius or Kelvin in lines 183-185.
Answer: We have corrected the units K or oC were they appeared misplaced.
- It is better to present a schematic figure for the two-step process.
Answer: After careful reading of the manuscript, we believe it flows and allows the reader to understand the content of the work. Thus, we have decided to keep the paper the way it is now, which helps to avoid adding more figures.
- Check references on lines 243, 264, 276, and more… Line 366 – Carvalho et al. citation style.
Answer: The format of the references has been corrected.
- There was no discussion in the SEM part. And it was similar in many other parts.
Answer: The manuscript was amended to include the torrefied vs. non-torrefied SEM imagery. Section 3.1 contains a short discussion on SEM results. We believe these are enough in the context of the paper.
- Why didn't you present the FTIR spectrum range between 650 to 4000 (Figure 2) when giving data that is not in the figure in the text (eg 2800-3000 and 700-900)?
Answer: Vibrations at 1700-1740 cm-1, 1250–1220 cm−1, 1269 cm−1, 1516 cm−1,1508 cm−1, 1600 cm-1, 1030-1100 cm-1 are represented by the provided FTIR spectra. “Data not shown” was used to designate vibrations at 700-900 cm-1 (800 cm-1), 3060–3100 cm−1, 2946-2987 cm-1, 2888-2946 cm-1, and 3200-3500 cm-1.
- Section 3.3 should be expanded.
Answer: We have accordingly expanded Section 3.3 to further discuss the organic carbon content in the torrefaction condensate and approach to maximizing it.
- In general, all references should be rechecked as some are missing or out of journal style.
Answer: As in our responses to points 2, 3, and 6, the missing references have been added and the format has been edited.
- The discussion part was too weak, it should be revised.
Answer: Thank you. We have accordingly revised the discussion part by rearranging or adding new text for clarification of torrefaction and anaerobic digestion.
- The aim of the article and the focused results were completely different, the article should be reviewed and re-evaluated.
Answer: We have modified the introductory abstract, title, conclusion accordingly to convey more consistency between focused results and aim of the article.
- The introduction part was too long.
Answer: We absolutely agree that Introduction part is long and extensive. However, we also humbly believe the content in this section is required for a better understanding of the work’s context and content and for adequately and comprehensively describing the complex integration of multiple biorefinery processes. Thus, only minor edits have been conducted in this section.

Round 2
Reviewer 4 Report
- Introduction of the abstract was too long.
- In the abstract, there was a typo (fractionation of wastewas here?)
- What is “TAG” in the abstract? Don’t use any abbreviations.
- The SEM figures (Figure 1a and 1b) should be the same size.
- In FTIR, the wavelength (12000) was wrong.
- In FTIR, the wavelength of 2800-3000 was not given. Its not good to not share the data if the band is decreasing at these range. Similarly, 700-900 was not given. This is very important data if aromatic concentrations are increasing.
- Moreover, its better to merge both FTIR figures (Figure 2 - raw and 280 ÌŠC-torrefied). This separated style is not scientific.
- There is also no TGA data. How we can evaluate the data similar to FTIR results?
- What are the concentrations of compounds (Table 1)? Levels of some compounds (maybe Phenol, 2,6-dimethoxy?) were dramatically increased, however it’s not seen easily on the graph. You should discuss them.
- Lactic acid can be deleted from Table 3, given the peak area that is not related to concentration.
- Table 4 is not clear, why do you have two different tables there. And why the concentrations are so different. Why carboxylic acid concentrations were increased if the second one is de-toxified?
- Check the reference list and revise them according to MDPI reference format. Don’t use the “et al.” in the reference list write the all of the authors. And References are not current, use the latest published papers (2018-2022).
Author Response
Response to Reviewer Comments and Suggestions
Thank you again for your feedback and for the opportunity to improve our scientific manuscript. Please also see the attachments for manuscript revision.
1)Introduction of the abstract was too long.
We removed the introduction of the abstract and moved it to the Introduction section.
2)In the abstract, there was a typo (fractionation of waste was here?)
Thank you. We inserted a space in this sentence to fix this typographical error.
3)What is “TAG” in the abstract? Don’t use any abbreviations.
We appropriately defined the acronym TAG as triacylgrlycerides.
4)The SEM figures (Figure 1a and 1b) should be the same size.
We resized the comparative SEM figures 1a and 1b to be the same size.
5)In FTIR, the wavelength (12000) was wrong.
We fixed the following general introductory statement that was cited from a peer-reviewed scientific journal article: “The infrared light has wavenumber range of 12800 ~ 10 cm−1[58]”.
6)In FTIR, the wavelength of 2800-3000 was not given. Its not good to not share the data if the band is decreasing at these range. Similarly, 700-900 was not given. This is very important data if aromatic concentrations are increasing.
We changed Figure 2 to depict overlapping FTIR spectra to more clearly depict changes between raw and torrefied wheat straw biomass. Figure 2 depicts changes in range of 800-900cm-1 described in text. We removed supplementary text relating to discussion of FTIR spectral range 2800-3000cm-1 and others that are not graphically depicted.
7)Moreover, its better to merge both FTIR figures (Figure 2 - raw and 280 ÌŠC-torrefied). This separated style is not scientific.
We merged both FTIR figures (Figure 2 - raw and 280 ÌŠC-torrefied)
8)There is also no TGA data. How we can evaluate the data similar to FTIR results?
We removed the erroneous and misleading statement “data not shown” to more clearly emphasize that the TGA data we obtained is used and represented by Figure 3. We state in the Materials and Methods Section that the isoconversional Friedman method [41], which assumes that the reaction rate (dα/dt) at a constant conversion is only a function of temperature, was applied to thermogravimetric data.
9)What are the concentrations of compounds (Table 1)? Levels of some compounds (maybe Phenol, 2,6-dimethoxy?) were dramatically increased, however it’s not seen easily on the graph. You should discuss them.
As was done for Figure 2 FTIR spectra, we modified Figure 4 by overlapping raw vs. torrefied py-GC-MS spectra to more clearly show the aforementioned changes in levels of compounds, and this is included in attached Supplementary Information 1 document. To better explain what the concentrations and relative abundance of compounds are in Table 1, we added the following text for discussion:
“Pyrolysis-gas chromatography–mass spectrometry (Py-GC–MS) is a technique which thermally decomposes a sample's large high-molecular weight molecules (at > 600°C in the absence of O2) into smaller and more volatile low-molecular weight fragments that are then separated by gas chromatography in an inert atmosphere or a vacuum depending on their volatility. Higher volatile particles travel faster through the column than lower volatile particles. The volatile molecules are then ionized using an electric charge and then sent to an electromagnetic field that filters the ions based on their specific mass (mass divided by charge number) expressed in units of m/z, which an ion detector will count. A mass spectrum correspondingly depicts the ion abundance that is counted by a detector via measurement of the current of electrons generated when the ions strike the detector for each m/z as a function of retention time. Spectra from Py-GC MS of raw and 300 °C-torrefied wheat straw revealed a marked change in relative peak areas for various compounds [Figure 4 and Supplementary Information 1]. These primary peaks corresponded to various compounds whose retention times are presented in Table 1.”
10)Lactic acid can be deleted from Table 3, given the peak area that is not related to concentration.
Lactic acid was deleted from Table 3.
11)Table 4 is not clear, why do you have two different tables there. And why the concentrations are so different. Why carboxylic acid concentrations were increased if the second one is de-toxified?
The second table depicted carboxylic acid content in purified, de-toxified liquid condensate arising from torrefaction of wheat-straw at 280 °C but was mistakenly not labelled with a Table ID# or caption. Concentrations in these two tables were different due to difference in measured ethanol and carboxylic content in liquid condensate derived from torrefaction of wheat straw at 280C and slightly higher 300C temperature prior to its purification and detoxification. For clarity, this second table was removed.
12. Check the reference list and revise them according to MDPI reference format. Don’t use the “et al.” in the reference list write the all of the authors. And References are not current, use the latest published papers (2018-2022).
We checked the reference list and revised them according to MDPI reference format. We removed "et al." in the reference list and included all authors and/or editors. These 58 citations informed the writing of the manuscript. The following key and current (2018-2022) published references were added during the editing process:
Yu, B.; Fan, G.; Zhao, S.; Lu, Y.; He, Q.; Cheng, Q.; Yan, J.; Chai, B.; Song, G. Yu, B. et al. Simultaneous isolation of cellulose and lignin from wheat straw and catalytic conversion to valuable chemical products. Applied Biological Chemistry, 2021, 64(15), 1-13; DOI: 10.1186/s13765-020-00579-x.
Hoang, A.T.; Nizetic, S.; Ong, H. C.; Chong, C.T.; Atabani, A.E.; Pham, V.V.; Hoang, A.T. Acid-based lignocellulosic biomass biorefinery for bioenergy production: Advantages, application constraints, and perspectives. Journal of Environmental Management, 2021, 296,1-24; DOI: 10.1016/j.jenvman.2021.113194.
Adhikari, S.; Nam, H.; Chakraborty, J.P. Chapter 8 - Conversion of Solid Wastes to Fuels and Chemicals Through Pyrolysis. Waste Biorefinery-Potential and Perspectives, 2018, 239-263; DOI:10.1016/B978-0-444-63992-9.00008-2.
Zhang,Y.; Xia,C.; Lu, M.; Tu, M. Yu, Z. Effect of overliming and activated carbon detoxification on inhibitors removal and butanol fermentation of poplar prehydrolysates. Biotechnology for Biofuels, 2018, 11(178),1-14; DOE: 10.1186/s13068-018-1182-0.
Chiranjeevi, T.; Mattam, A.J.; Vishwakarma, K.K.; Uma, A.; Peddy, V.C.R.; Gandham, S.; Velankar, H.R. Chiranjeevi, T. Assisted Single-Step Acid Pretreatment Process for Enhanced Delignification of Rice Straw for Bioethanol Production. ACS Sustainable Chemical Engineering, 2018, 6(7), 8762-8774; DOI: 10.1021/acssuschemeng.8b01113.
Mamvura, T.A.; Danha, G. Biomass torrefaction as an emerging technology to aid in energy production. Heliyon, 2020, 6(3),1-17; DOI: 10.1016/j.heliyon. 2020.e03531.
